# Metabolic control of cellular immune-competency by odors in *Drosophila*

**Sukanya Madhwal[1,2], Mingyu Shin[3], Ankita Kapoor[1,2], Manisha Goyal[1,4], Manish K Joshi[1†], Pirzada Mujeeb Ur Rehman[1‡], Kavan Gor[1], Jiwon Shim[3,5]\*, Tina Mukherjee[1]\***

[1]Institute for Stem Cell Science and Regenerative Medicine (inStem), Bangalore, India; [2]Manipal Academy of Higher Education, Manipal, India; [3]Department of Life Science, College of Natural Science, Hanyang University, Seoul, Republic of Korea; [4]The University of Trans-Disciplinary Health Sciences & Technology (TDU), Bengaluru, India; [5]Research Institute for Natural Science, Hanyang University, Seoul, Republic of Korea

**Abstract** Studies in different animal model systems have revealed the impact of odors on immune cells; however, any understanding on why and how odors control cellular immunity remained unclear. We find that *Drosophila* employ an olfactory-immune cross-talk to tune a specific cell type, the lamellocytes, from hematopoietic-progenitor cells. We show that neuronally released GABA derived upon olfactory stimulation is utilized by blood-progenitor cells as a metabolite and through its catabolism, these cells stabilize Sima/HIFα protein. Sima capacitates blood-progenitor cells with the ability to initiate lamellocyte differentiation. This systemic axis becomes relevant for larvae dwelling in wasp-infested environments where chances of infection are high. By co-opting the olfactory route, the preconditioned animals elevate their systemic GABA levels leading to the upregulation of blood-progenitor cell Sima expression. This elevates their immune-potential and primes them to respond rapidly when infected with parasitic wasps. The present work highlights the importance of the olfaction in immunity and shows how odor detection during animal development is utilized to establish a long-range axis in the control of blood-progenitor competency and immune-priming.

**\*For correspondence:**
jshim@hanyang.ac.kr (JS);
tinam@instem.res.in (TM)

**Present address:** †Aix Marseille Université, CNRS, Institut de Biologie du Développement de Marseille (IBDM), Marseille, France; ‡University of Cologne, CECAD-Cluster of Excellence, Köln, Germany

**Competing interests:** The authors declare that no competing interests exist.

## Introduction

Hematopoiesis in *Drosophila* gives rise to three blood cell types: plasmatocytes, crystal cells, and lamellocytes, with characteristics that are reminiscent of the vertebrate myeloid lineage. Of these, lamellocytes which are undetectable in healthy animals, appear upon infections with the parasitic wasp, *Leptopilina boulardi (L. boulardi)* which triggers their development (*Crozatier et al., 2004*). Within a few hours of wasp-egg deposition, the *Drosophila* larval hematopoietic system activates a series of cellular innate immune responses leading to massive differentiation of blood cells into lamellocytes. This includes trans-differentiation of circulating and sessile plasmatocytes and differentiation of multipotent blood-progenitor cells of the larval hematopoietic organ termed the 'lymph gland' (*Anderl et al., 2016*; *Honti et al., 2010*; *Márkus et al., 2009*; *Stofanko et al., 2010*). As lymph gland progenitor cells differentiate, the gland ultimately disintegrates to release its blood cells into circulation (*Lanot et al., 2001*). Together, these events contribute toward robust lamellocyte numbers which reach a maximum at 48 hr after wasp-egg laying (*Lanot et al., 2001*). Characterized by their large flattened appearance, lamellocytes encapsulate the deposited wasp-eggs and melanize them, facilitating their effective clearance (*Rizki and Rizki, 1992*). Lamellocyte differentiation is controlled by signals of both local and systemic origin. They encompass autonomous cell-fate-determining programs (*Dragojlovic-Munther and Martinez-Agosto, 2012*; *Sinenko et al., 2011*;

*Makki et al., 2010*; *Small et al., 2014*) and global metabolic adaptation processes that are initiated upon infection (*Bajgar et al., 2015*; *Dolezal et al., 2019*). Blood cells therefore maintain a demand – adapted hematopoietic process to develop lamellocytes. This innate competitiveness provides a defence mechanism for the fly to limit parasitoid success. An understanding of developmental programs that prime immune-progenitor cells with potential to respond when in need forms the central focus of this investigation.

Development of multipotent blood-progenitor cells of the lymph gland relies on cues of autonomous (*Benmimoun et al., 2012*; *Krzemień et al., 2007*)and non-autonomous origin (*Banerjee et al., 2019*; *Morin-Poulard et al., 2016*). Of these, olfactory signaling has been implicated in their maintenance (*Shim et al., 2013*). Interestingly, studies in different model systems have revealed the impact of odors on immune cells (*Strous and Shoenfeld, 2006*), and revealed the influence of odors and their specificity in mediating cellular responses. Any understanding on why and how odors control cellular immunity, however, remains unclear. The present work highlights the importance of the olfaction/immune axis in immunity.

The *Drosophila* larval olfactory system contains 25 specific odorant receptors (OR) in 21 olfactory receptor neurons (ORNs). Orco (Or83b), an atypical OR protein, expressed in every ORN is necessary to respond to all odors. Odors are sensed by larval dorsal organ, which is innervated by dendrites of these ORNs that project to specific glomerulus of the larval antennal lobe. Here, ORNs form excitatory synapses with projection neurons (PN) whose axons innervate into regions of the brain representative of higher order information processing. The different glomeruli are interconnected by excitatory or inhibitory local interneurons that fine-tune the ORN-PN network. It has been previously shown that during *Drosophila* larval development, olfaction stimulates the release of GABA from neurosecretory cells of the brain, which systemically activates GABA$_B$R signaling in the progenitor cells to support their maintenance (*Shim et al., 2013*). In animals with olfactory dysfunction, this systemic cross-talk is perturbed and drives precocious differentiation of blood-progenitor cells. In the current study, we show that animals employ the olfactory/immune cross-talk to tune lamellocyte potential of hematopoietic-progenitor cells. The neuronally released GABA derived upon olfactory stimulation is utilized by blood-progenitor cells as a metabolite to stabilize Sima/HIFα protein. Sima is a well-characterized transcription factor known for its role in inducing hypoxia response (*Romero et al., 2007*; *Semenza and Wang, 1992*), which in immune progenitor cells is necessary to drive lamellocyte differentiation. While developmentally the olfaction/GABA systemic axis sustains the ability of progenitor cells to differentiate into lamellocytes, animals rearing in parasitoid threatened states co-opt this olfactory axis to prime immunity to respond to infections more rapidly and effectively. Overall, our study explores the mechanistic and physiological relevance of the olfaction/immune connection during *Drosophila* larval hematopoiesis and establishes its importance in the maintenance of a competent demand-adapted immune system.

## Results

### Olfaction controls cellular immune response necessary to combat parasitic wasp infections

In order to assess the influence of olfaction on the immunity, we infected *Drosophila* larvae with olfactory dysfunction, with parasitic wasps, *L. boulardi* and assessed for cellular immune response. We analyzed: (1) *orco* mutant (*Neuhaus et al., 2005*), the common odorant co-receptor 83b necessary for all odor responsiveness (*Larsson et al., 2004*) (*orco¹/orco¹*; *Figure 1A–D*), (2) *Orco>Hid, rpr*, which genetically ablated all ORNs (*Figure 1—figure supplement 1A,B*), and (3) *Or42a>Hid*, which specifically ablated Or42a, the ORN implicated in sensing of food-related odors (*Figure 1E– G*). We addressed the cellular immune response to wasp-infection in genetic and physiological conditions with altered odor environment. Lamellocytes were assessed in the lymph gland at 24 hr post-infection (HPI) (*Figure 1A*), and in circulation at 48 HPI (*Figure 1A'*). A comparative analysis of immune cells was also undertaken in uninfected conditions both in the lymph gland and circulation (shown in *Supplementary files 1* and *2*). We found that animals with olfactory dysfunction demonstrated a specific loss in lamellocyte formation (*Figure 1A–G*, *Figure 1—figure supplement 1A–E*). The formation of mature immune cell types seen in homeostasis like the crystal cells (*Figure 1—figure supplement 1F*) and plasmatocytes (*Shim et al., 2013*), however, remained unaffected. This

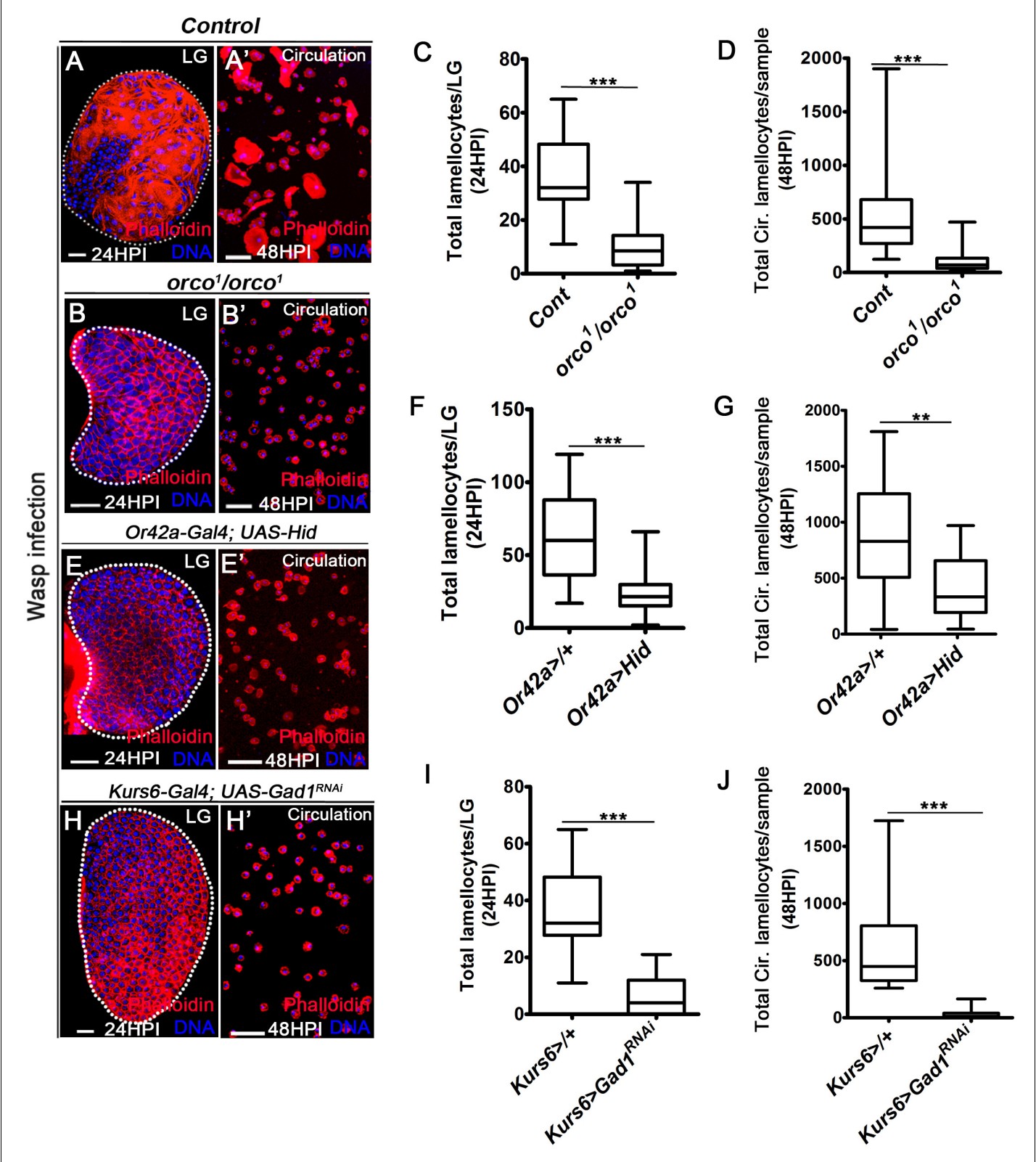

**Figure 1.** Odor-mediated neuronal GABA availability specifies lamellocyte potential. DNA is stained with DAPI (blue). Phalloidin (red) marks blood cells and lamellocytes are characterized by their large flattened morphology. Scale bars in panels A, B, E, H = 20 μm and A', B', E', H' = 50 μm. HPI indicates *hours post wasp-infection*, and LG is lymph gland. In lymph gland, lamellocytes analyzed at 24HPI and in circulation at 48HPI. In panels (C, D, F, G, I, J), median is shown in box plots and vertical bars represent upper and lowest cell-counts and statistical analysis is Mann-Whitney test, two-

*Figure 1 continued on next page*

*Figure 1 continued*

tailed. 'n' represents the total number of larvae analyzed, and for lymph gland 'n' represents lymph gland lobes analyzed. White dotted lines demarcate lymph glands. For better representation of the lymph gland primary lobes, the images shown, have been edited for removal of adjacent tissues (like dorsal vessel and ring gland). (A–A') Control ($w^{1118}$) infected larvae showing lamellocyte induction in (A) lymph gland and (A') circulation. (B–D) Compared to (A–A') control ($w^{1118}$), $orco^1/orco^1$ mutant larvae show reduction in lamellocyte in (B, C) lymph gland (n = 20, ***p<0.0001 compared to $w^{1118}$, n = 18) and (B', D) circulation (n = 18, ***p<0.0001 compared to $w^{1118}$, n = 23). (E–G) Specifically ablating Or42a (Or42a-Gal4; UAS-Hid) causes reduction in lamellocytes in (E, F) lymph gland (n = 24, ***p<0.0001 compared to Or42a-Gal4/+, n = 24) and (E', G) circulation (n = 20, **p=0.004 compared to Or42a-Gal4/+, n = 19). (H–J) Blocking neuronal GABA bio-synthesis in Kurs6$^+$ neurons (Kurs6-Gal4; UAS-Gad1$^{RNAi}$) recapitulates lamellocyte reduction in (H, I) lymph gland (n = 22, ***p<0.0001 compared to Kurs6-Gal4/+, n = 18) and (H', J) circulation (n = 25, ***p<0.0001 compared to Kurs6-Gal4/+, n = 22).

The online version of this article includes the following source data and figure supplement(s) for figure 1:

**Source data 1.** Contains numerical data plotted in *Figure 1C,D,F,G,I and J*.
**Figure supplement 1.** Olfaction/GABA axis controls lamellocyte induction.
**Figure supplement 1—source data 1.** Contains numerical data plotted in *Figure 1—figure supplement 1A,B,C,D,E,F and G*.
**Figure supplement 2.** Expression profile of the neuronal driver lines.
**Figure supplement 2—source data 1.** Contains numerical data plotted in *Figure 1—figure supplement 2E,J,O and T*.

implied a specific role for olfaction in controlling lamellocyte formation. Upon wasp-infection, apart from *Orco>Hid, rpr* where a reduction in total cell numbers was apparent in comparison to its control, all other genetic contexts showed comparable cell densities with respect to their stage matched control (*Figure 1—figure supplement 1E*). The reduction in *Orco>Hid, rpr* seemed specific to *Orco>* background as any such change in *orco* mutants was undetectable. These data implied that the lamellocyte defect was not a consequence of general dampening of immune cell numbers and was specific to the loss of lamellocyte potential in olfactory mutants. In uninfected conditions, analysis of lymph gland and circulating hemocytes for lamellocytes and immune cell densities in olfactory mutant genetic contexts did not reveal any difference and were comparable to controls (*Supplementary files 1* and *2*). These data strengthened the importance of olfaction in wasp-infection-mediated lamellocyte response.

The genetic perturbation of Or42a suggested a specific requirement of food-odor sensing in lamellocyte development (*Figure 1E–G*). This was further supported by a physiological experimental set up designed to test the involvement of food odors in cellular immune response toward infection (*Figure 1—figure supplement 1C,D*). For this, *Drosophila* larvae were reared from early embryonic stage in food medium with minimal odors but nutritionally equivalent to regular diet (*Shim et al., 2013*). These animals were then infected with wasps and their lamellocyte response was analyzed. Interestingly, this condition recapitulated the lamellocyte formation defect seen upon loss of Or42a. Supplementing the minimal medium with food odors corrected the defect (*Figure 1—figure supplement 1C,D*). Together, with the loss-of-function mutation and food-odor experiment, the data demonstrated the importance of food odors and their sensing in lamellocyte cell fate specification.

Downstream of odor detection, activation of PN is necessary to mediate systemic release of GABA from neurosecretory cells of the larval brain. The GABA-producing neurosecretory cells are marked with *Kurs6-Gal4* driver (Kurs6$^+$)-based expression of reporter transgenes (*Shim et al., 2013*). Blocking PN signaling (GH146>ChAT$^{RNAi}$) abrogated lamellocyte formation (*Figure 1—figure supplement 1G*). Similarly, blocking GABA biosynthesis in GABA-producing neurosecretory cells (Kurs6>Gad1$^{RNAi}$) led to specific loss in lamellocyte formation in response to infection (*Figure 1H–J*, *Figure 1—figure supplement 1E*). This genetic condition did not impede differentiation into crystal cells (*Figure 1—figure supplement 1F*) or formation of plasmatocytes (*Shim et al., 2013*), in homeostatic (uninfected) conditions. Expression of the aforementioned neuronal driver lines is limited to the nervous system (*Figure 1—figure supplement 2A,B,F,G,K,L,P and Q*; *Shim et al., 2013*). Therefore, the lamellocyte phenotypes detected, report genetic manipulations in the neuronal tissue and are not a consequence of nonspecific expression of the drivers in the lymph gland in the conditions tested (*Figure 1—figure supplement 2C,D,H,I,M,N,R and S*). Moreover, the above-mentioned neuronal manipulations did not affect PSC (posterior signaling center, niche) cell numbers, whose function in cellular immune response has been well established (*Makki et al., 2010*; *Louradour et al., 2017*; *Figure 1—figure supplement 2E,J,O and T*). Hence, these data revealed a

specific role for olfactory stimulation-dependent, downstream PN signaling and neuronally derived GABA, in priming immune cells with lamellocyte potential.

Lamellocytes are derived from both circulating pool of immune cells and also from multipotent-progenitor cells of the lymph gland (*Louradour et al., 2017*; *Sorrentino et al., 2002*). Olfaction has been shown to control maintenance of lymph gland progenitor cells through the systemic use of neuronally derived GABA (*Shim et al., 2013*). The accompanying lamellocyte defect detected within the lymph gland samples of the olfactory and neuronal mutant animals, led us to investigate the mechanistic underpinnings of this systemic axis on lamellocyte differentiaion within the lymph gland blood progenitor-cells.

## GABA uptake and metabolism in blood-progenitor cells controls lamellocyte formation

Neuronally derived GABA activates GABA$_B$R/Ca$^{2+}$-CaMKII signaling in blood-progenitor cells of the lymph gland (*Shim et al., 2013*). Hence, we reasoned a role for GABA$_B$R function in progenitor cells and examined lamellocyte formation upon progenitor-specific loss of *GABA$_B$R1*, achieved by expressing *GABA$_B$R1$^{RNAi}$* using progenitor-specific drivers, *dome-MESO-Gal4* and *Tep4-Gal4*. Both the driver lines in 2nd and 3rd instar larval lymph glands showed restricted expression within blood-progenitor cells of uninfected and infected animals and not in the cells of PSC (*Makki et al., 2010*; *Figure 2—figure supplement 1A–H', I and J*). In an analysis of 2nd and 3rd instar larval circulating blood cells in uninfected and infected conditions, *dome-MESO-Gal4* expression was minimally detected in circulating immune cells (*Figure 2—figure supplement 1A''-C''*). However, at 24HPI, *dome-MESO-Gal4* expression was detected in circulating blood cells as well (*Figure 2—figure supplement 1D''*). *Tep4-Gal4* expression was undetectable in circulation (*Figure 2—figure supplement 1E'-H''*).

Abrogating GABA$_B$R function in progenitor cells did not impede lamellocyte development in response to wasp-infection. A significant increase in lymph gland lamellocyte numbers was evident at 24HPI (*Figure 2—figure supplement 1K*). Their numbers in circulation at 48HPI, however, remained comparable to control genetic backgrounds (*Figure 2—figure supplement 1L,N and O*). The formation of a few lamellocytes could be detected in *GABA$_B$R1$^{RNAi}$* expressing animals in uninfected conditions (*Supplementary file 1* and *2*). Together, these data implied that progenitor loss of *GABA$_B$R1* did not affect lamellocyte formation. The overall cellular response to infection also remained unaffected (*Figure 2—figure supplement 1M* and *Supplementary file 1* and *2*). pCaMKII expression, a downstream read out of GABA$_B$R signaling was also analyzed in lymph glands obtained from animals post wasp-infection which remained unchanged (*Figure 2—figure supplement 1P–R*). These data implied a GABA$_B$R-signaling-independent function for GABA in lamellocyte differentiation. The mechanism by which neuronally derived GABA influenced blood-progenitor cell differentiation into lamellocytes was explored next.

Independent of activating GABA$_B$R signaling, GABA function as a metabolite is well described (*Bouché and Fromm, 2004*; *Shelp et al., 1999*; *Maguire et al., 2015*). This led us to explore the metabolic implications of GABA in the immune response. This was undertaken by an expression analysis of Gat, a functional GABA-transporter that facilitates GABA uptake (*Figure 2A*, *Thimgan et al., 2006*) and analysis of intracellular GABA levels (iGABA, see methods for staining details) in lymph gland tissues. In homeostatic conditions, Gat expression, using an anti-Gat antibody (*Muthukumar et al., 2014*), revealed uniform levels in all cells of the lymph gland (*Figure 2B*). Within 6 hr of wasp-infection (6HPI), a twofold upregulation was detected in all blood cells of the lymph gland (*Figure 2C,D*). Correspondingly, at 6HPI a twofold increase in iGABA levels was also noticed (*Figure 2E–G*). The iGABA levels were sensitive to changes in blood-progenitor *Gat* expression (*Figure 2H–K*). Downregulating *Gat* in blood-progenitor cells using *Gat$^{RNAi}$*, (*dome-MESO>Gat$^{RNAi}$*) was sufficient to substantially reduce iGABA levels in homeostasis (*Figure 2H,J*) and post-infection (*Figure 2I,K*). This suggested a role for Gat in moderating intracellular GABA levels in blood-progenitor cells. Downregulating *Gat* (*dome-MESO>Gat$^{RNAi}$* and *Tep4>Gat$^{RNAi}$*) resulted in a dramatic loss in lamellocyte formation in the lymph gland (*Figure 2L,M and P* and *Figure 2—figure supplement 2A,J*) and circulation (*Figure 2Q* and *Figure 2—figure supplement 2B,E,F and K*). Using additional RNA*i* lines we corroborated the lamellocyte phenotype, and observed a reduction at 48HPI in circulation (*Figure 2—figure supplement 2A,B*). On the other hand, over-expression of *Gat* in progenitor cells (*dome-MESO>Gat*), which elevated intracellular GABA levels in lymph gland blood cells

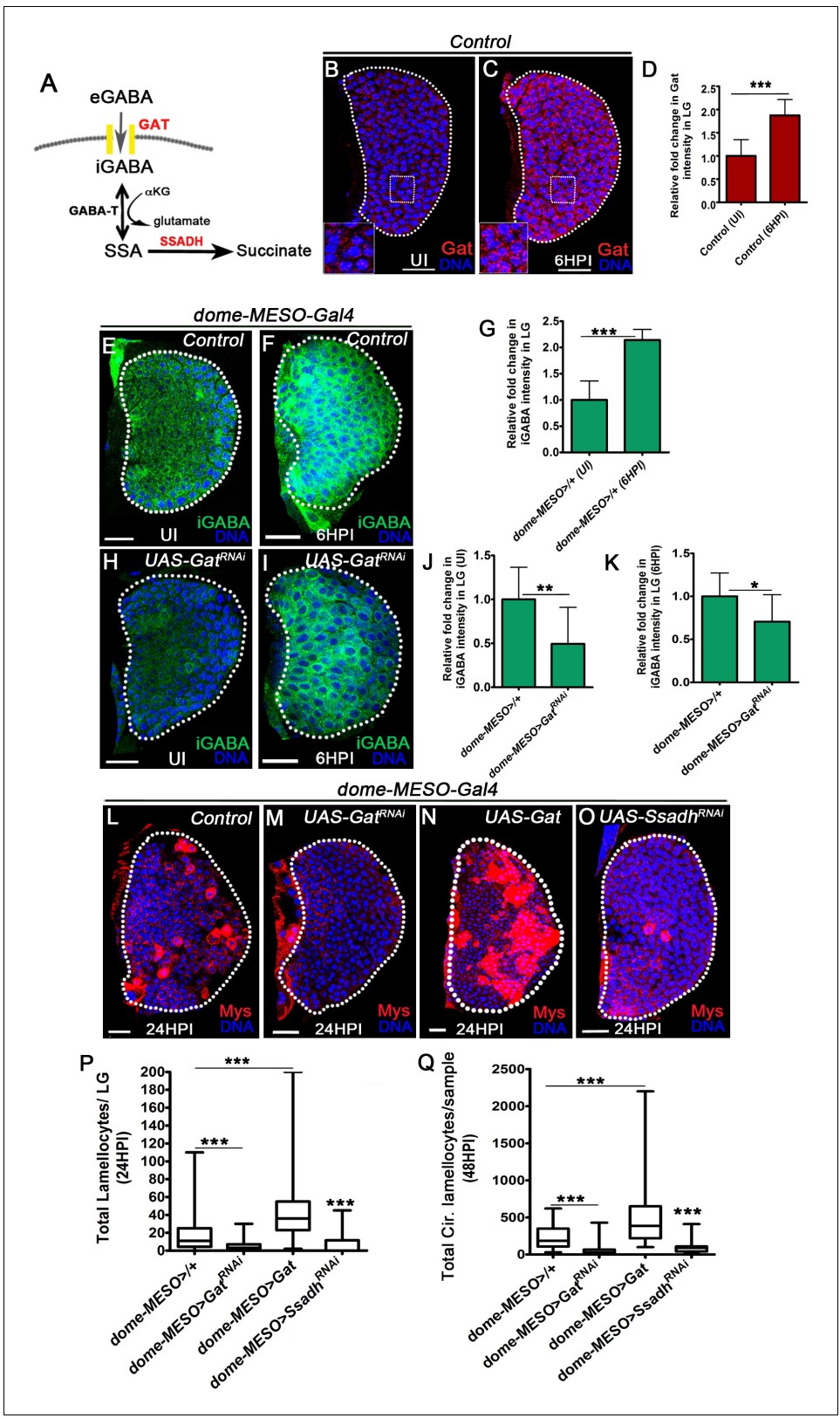

**Figure 2.** GABA-uptake and catabolism is necessary for lamellocyte formation. DNA is marked with DAPI (blue), GABA Transporter (Gat, red), intracellular GABA (iGABA, green), and blood cells are marked with phalloidin (red). Myospheroid (Mys) antibody staining (in panels, (**L–O**), red) is employed to mark the lamellocytes in lymph gland. Scale bars in panels (**B, C, E, F, H, I, L-O** = 20 µm). UI indicates uninfected, HPI indicates *hours post-wasp-infection*. In lymph gland, lamellocytes analyzed at 24HPI and in circulation at 48HPI. In panels (**D, G, J** and **K**), mean with standard deviation

*Figure 2 continued on next page*

*Figure 2 continued*

is shown and in panels (**P** and **Q**), median is shown in the box plots and vertical bars represent upper and lowest cell counts. Statistical analysis applied in panels (**D, G, J,** and **K**) is unpaired *t*-test, two-tailed and in panels (**P** and **Q**) is Mann-Whitney test, two-tailed. 'n' is the total number of larvae analyzed, and for lymph gland 'n' represents lymph gland lobes analyzed. White dotted lines demarcate lymph glands. For better representation of the lymph gland primary lobes, the images shown, have been edited for removal of adjacent tissues (like dorsal vessel and ring gland). (**A**) Schematic of the GABA-shunt pathway. Uptake of extra cellular GABA (eGABA) via GAT (yellow bars) in blood-progenitor cells and its intracellular catabolism through GABA-transaminase (GABA-T) which catalyzes the conversion of GABA into succinic semi-aldehyde (SSA) and its further breakdown into succinate by succinic semi-aldehyde dehydrogenase (SSADH, rate limiting step). (**B, C**) Uniform GABA transporter (Gat) expression is detected in control (*Hml^△-Gal4, UAS-GFP*) lymph gland from (**B**) Uninfected *Drosophila* larvae. In comparison to uninfected lymph gland, (**C**) Gat expression is elevated in lymph gland at 6HPI. Inset in both **A, B** shows zoomed image for the selected region in lymph gland for better clarity. Scale bars correspond to the main lymph gland image and not the inset. See corresponding quantifications in (**D**). (**D**) Relative fold change in Gat expression in control lymph gland from uninfected and infected states at 6HPI. Compared to uninfected control lymph gland (*Hml^△-Gal4, UAS-GFP/+*, n = 15), almost twofold increase in Gat expression is observed at 6HPI (*Hml^△-Gal4, UAS-GFP/+*, n = 16, \*\*\*p<0.0001). (**E, F**) Control (*dome-MESO>GFP/+*) lymph glands from (**E**) Uninfected *Drosophila* larvae show punctated iGABA staining in all blood cells (anti-GABA antibody staining in 1X PBS + 0.3%Triton X-100). In comparison to (**E**), iGABA levels detected in (**F**) lymph gland at 6HPI is elevated. See corresponding quantifications in (**G**). (**G**) Relative fold change in iGABA levels in uninfected and infected control lymph glands at 6HPI. Compared to uninfected control lymph gland (*dome-MESO-Gal4, UAS-GFP/+*, n = 9), a twofold increase in iGABA expression is observed at 6HPI (*dome-MESO-Gal4, UAS-GFP/+*, n = 11, \*\*\*p<0.0001). (**H, I**) Loss of progenitor *Gat* function (*dome-MESO-Gal4, UAS-GFP; UAS-Gat^RNAi*) leads to reduced iGABA levels both in (**H**) uninfected and (**I**) infected states as compared to control (*dome-MESO-Gal4, UAS-GFP/+*) in (**E**) uninfected and (**F**) infected states, respectively. See corresponding quantifications in (**J** and **K**). (**J**) Relative fold change in iGABA levels in lymph glands from uninfected *dome-MESO-Gal4, UAS-GFP/+* (control, n = 10) and *dome-MESO-Gal4, UAS-GFP; UAS-Gat^RNAi* (n = 10, \*\*p=0.0097). (**K**) Relative fold change in iGABA expression in lymph glands at 6HPI in *dome-MESO-Gal4, UAS-GFP/+* (control, n = 11) and *dome-MESO-Gal4, UAS-GFP; UAS-Gat^RNAi* (n = 11, \*p=0.0286). (**L–O**) In response to wasp-infection, lamellocytes detected in (**L**) Control lymph gland. (**M**) Expressing *Gat^RNAi* in progenitor cells (*dome-MESO-Gal4, UAS-GFP; UAS-Gat^RNAi*) causes reduction in lamellocyte numbers in lymph gland. However, (**N**) Expressing *Gat* in progenitor cells (*dome-MESO-Gal4, UAS-GFP; UAS-Gat*) leads to increased number of lamellocytes in lymph gland. (**O**) Expressing *Ssadh^RNAi* in progenitor cells (*dome-MESO-Gal4, UAS-GFP; UAS-Ssadh^RNAi*) causes reduction in lamellocyte numbers in lymph gland compared to (**L**) Control lymph gland response. See corresponding quantifications in (**P**). (**P**) Quantifications of progenitor-specific knock-down of *Gat*, over-expression of *Gat* and *Ssadh* knock-down showing lymph gland lamellocyte numbers, *dome-MESO-Gal4, UAS-GFP/+* (control, n = 68), *dome-MESO-Gal4, UAS-GFP; UAS-Gat^RNAi* (n = 42, \*\*\*p<0.0001), *dome-MESO-Gal4, UAS-GFP; UAS-Gat* (n = 63, \*\*\*p<0.0001), *dome-MESO-Gal4, UAS-GFP; UAS-Ssadh^RNAi* (n = 57, \*\*\*p=0.0001). (**Q**) Quantifications of progenitor-specific knock-down of *Gat*, over-expression of *Gat* and *Ssadh* knock-down showing lamellocytes numbers in circulation, *dome-MESO-Gal4, UAS-GFP/+* (control, n = 42), *dome-MESO-Gal4, UAS-GFP; UAS-Gat^RNAi* (n = 37, \*\*\*p<0.0001), *dome-MESO-Gal4, UAS-GFP; UAS-Gat* (n = 40, \*\*\*p<0.0001) and *dome-MESO-Gal4, UAS-GFP; UAS-Ssadh^RNAi* (n = 22, \*\*\*p<0.0001). The online version of this article includes the following source data and figure supplement(s) for figure 2:

**Source data 1.** Contains numerical data plotted in *Figure 2D,G,J,K,P and Q*.
**Figure supplement 1.** GABA receptor signaling is not required for lamellocyte formation.
**Figure supplement 1—source data 1.** Contains numerical data plotted in *Figure 2—figure supplement 1I,J,K,L,M,N,O and R*.
**Figure supplement 2.** GABA uptake and its metabolism is important for lamellocyte formation.
**Figure supplement 2—source data 1.** Contains numerical data plotted in *Figure 2—figure supplement 2A,B,I,J,K and M*.
**Figure supplement 3.** GABA-shunt pathway is dispensable for normal hematopoiesis.
**Figure supplement 3—source data 1.** Contains numerical data plotted in *Figure 2—figure supplement 3P,Q and R*.
**Figure supplement 4.** GABA-shunt-derived succinate controls lamellocyte potential.
**Figure supplement 4—source data 1.** Contains numerical data plotted in *Figure 2—figure supplement 4C,I and J*.
**Figure supplement 5.** Lamellocyte induction is independent of TCA-cycle enzymes, *αKDH*, *Skap*, and *Sdh*.
**Figure supplement 5—source data 1.** Contains numerical data plotted in *Figure 2—figure supplement 5A,B,C and E*.

(*Figure 2—figure supplement 2C,D*) was sufficient to expand lamellocyte numbers both in lymph gland (*Figure 2N,P*) and circulation (*Figure 2Q* and *Figure 2—figure supplement 2G*). Unlike other genetic conditions, *Gat* over-expressing animals showed sporadic formation of lamellocytes even in uninfected conditions albeit at lesser numbers than seen in response to infection (*Supplementary file 1* and *2*). This showed that Gat function in progenitor cells was necessary and sufficient for lamellocyte determination. Gat expression in progenitor cells was limiting and raising its levels either genetically or upon infection led to expansion of lamellocyte numbers. These genetic perturbations did not alter blood-cell densities post-infection or in uninfected states (*Figure 2—figure supplement 2I* and *Supplementary file 2*), implying specificity in Gat function in controlling lamellocyte differentiation without affecting overall blood development.

To address the underlying cause for the lamellocyte defect seen in *Gat^RNAi*, we investigated the intracellular functions of GABA. Intracellularly, GABA can be catabolized via the GABA-shunt pathway to generate succinate in two steps (*Shelp et al., 1999*). The final step catalyzed by succinic-

semialdehyde dehydrogenase (Ssadh, *Figure 2A*) is the rate-limiting and critical step of the GABA-catabolic pathway (*Shelp et al., 1999*). Hence, we manipulated this step by expressing *Ssadh*^RNAi in blood-progenitor cells. Recapitulating the lamellocyte defect seen in olfactory and *Gat* loss-of-function conditions, loss of *Ssadh* in progenitor cells resulted in a lamellocyte reduction phenotype both in the lymph gland (*Figure 2O and P* and *Figure 2—figure supplement 2A and J*) and in circulation (*Figure 2Q* and *Figure 2—figure supplement 2B,H and K*). Again, the loss of progenitor *Ssadh* expression did not affect overall blood cell density, in development (*Supplementary file 1* and *2*) or in response to infection (*Figure 2—figure supplement 2I*). The requirement for *Ssadh* function in lamellocyte formation also correlated with its expression in lymph gland blood cells (*Figure 2—figure supplement 2L,M*). This was determined using in situ hybridization (*Figure 2—figure supplement 2L*) and quantitative real-time PCR (*Figure 2—figure supplement 2M*).

If the lack of lamellocyte formation in *Gat* or *Ssadh* loss-of-function condition was a consequence of aberrant lymph gland blood development was investigated (*Figure 2—figure supplement 3*). For this, hematopoiesis in homeostatic condition in *dome-MESO>Gat*^RNAi and *dome-MESO>Ssadh*^RNAi expressing lymph glands was assessed. Overall analysis of lymph gland development did not reveal dramatic changes in differentiation of progenitor population (measured by assessing DomeGFP^+ area), their maintenance (*Figure 2—figure supplement 3A–C*) or the proportion of differentiated mature blood cells (*Figure 2—figure supplement 3A–F,P and Q*, *Supplementary file 3* and *5*) except for a marginal increase in intermediate progenitor population co-expressing Dome and Pxn (Dome^+Pxn^+, *Figure 2—figure supplement 3P* and *Supplementary file 3*). This increase, however, did not lead to any increase in mature blood cells of crystal cells and plasmatocyte population; with numbers remained comparable to controls (*Figure 2—figure supplement 3A–F,P and Q*, *Supplementary file 3* and *Supplementary file 5*). Expression analysis of pCaMKII, Wingless, Ci^155 levels and PSC cell number, parameters implicated in progenitor homeostasis, did not reveal any changes in expression patterns or levels (*Figure 2—figure supplement 3G–O,R*). These data showed that Gat and Ssadh function in progenitor cells was largely dispensable for steady-state hematopoiesis, but these proteins were critical for demand-induced hematopoiesis in response to wasp-infections.

## GABA-catabolism-derived succinate is necessary for lamellocyte formation

The metabolic output of Ssadh enzymatic reaction is the generation of succinate (*Figure 2A*). Hence, we explored if supplementing succinate to *Drosophila* larvae expressing *Gat*^RNAi or *Ssadh*^RNAi in blood-progenitor cells corrected their lamellocyte defects. For this, synchronized first instar larvae were raised on food containing 3–5% succinate and then subjected to wasp-infections, which was followed by analysis of their cellular immune response. This diet did not affect general aspects of lymph gland development and hematopoiesis. Progenitor and differentiated blood-cell profiles showed no changes and remained comparable to larvae raised on regular food (*Figure 2—figure supplement 4A–C* and *Supplementary file 3*). Compared to *Gat*^RNAi or *Ssadh*^RNAi mutants raised on regular food, the succinate supplemented diet significantly restored lamellocyte numbers in response to infection (*Figure 2—figure supplement 4D–J*). This was evident both in lymph glands (*Figure 2—figure supplement 4D–H and I*) and circulating lamellocyte counts (*Figure 2—figure supplement 4D'–H' and J*). These data suggested an importance of GABA-catabolism-derived succinate in lamellocyte induction.

Succinate is also derived from the tricarboxylic acid cycle (TCA) via the conversion of α-ketoglutarate, which is catalyzed in a two-step process by *α-ketoglutarate dehydrogenase, αKDH* (*CG33791*) (*Zhou et al., 2008*) and *succinyl CoA synthetase, skap* (*CG11963*) (*Gao et al., 2008*). Downregulating these TCA enzymes did not lead to any defect in lamellocyte formation (*Figure 2—figure supplement 5A–C*). Even though the expression of these enzymes is detected in lymph glands (*Figure 2—figure supplement 5D and E*) their loss-of-function data highlighted a TCA-independent but GABA-catabolism-dependent control of lamellocyte differentiation in blood-progenitor cells. The independence of TCA in this context is intriguing, and we speculate separate pools of succinate in blood cells that are maintained to control basal cellular metabolism and specialized immune requirements. The TCA-derived succinate most likely conducts basal metabolic functions, and the GABA-catabolism-derived succinate sustains the immune requirement of these blood cells. As a result, blocking GABA uptake and its catabolism without compromising basal cellular metabolism

still allowed the development and differentiation to other blood cell types, but nevertheless impeded lamellocyte potential. Together, these data revealed a role for GABA-catabolic pathway in supporting succinate availability in blood cells upon wasp-infections that is necessary toward mounting a lamellocyte response.

## Lamellocyte differentiation is Sima dependent

We next explored the downstream effector role of succinate in lamellocyte differentiation. As a metabolite, succinate can fuel the activity of succinate dehydrogenase (SDH) complex, which is the complex II of the mitochondrial respiratory chain that converts succinate to fumarate (*Rutter et al., 2010*). We expressed RNAi against the catalytic subunit of Sdh (*SdhA^RNAi*) in progenitor cells as the means to inhibit SDH function and prevent succinate utilization within the TCA. This genetic perturbation failed to show any reduction in lamellocyte numbers (*Figure 2—figure supplement 5A–C*) and implied an alternative route for succinate function in lamellocyte formation.

Multiple studies across model systems and cell types have reported an integral role for succinate in hypoxia-independent stabilization of Hypoxia-inducible factor (HIF1α via inhibition of prolyl hydroxylases that mark HIF1α protein for degradation [*Brière et al., 2005*; *Selak et al., 2005*; *Tannahill et al., 2013*]). Within the *Drosophila* larval hematopoietic tissue, Sima protein, orthologous to mammalian HIF1α (*Lavista-Llanos et al., 2002*) is detected at basal levels in all cells of the larval lymph gland and cells of the PSC (*Figure 3A*, *Figure 3—figure supplement 1A,B*) with comparatively higher expression in crystal cells as previously reported (*Mukherjee et al., 2011*; *Figure 3A* and *Figure 3—figure supplement 1C–D'*). Within a few hours of wasp-infection (6HPI), a twofold upregulation in expression of Sima protein in lymph gland blood cells was noticed (*Figure 3B and E*). This was observed prior to detection of any lamellocyte formation. Later, when formation of lamellocyte was detected (12HPI and 24 HPI), Sima protein expression was seen in them as well (*Figure 3—figure supplement 1E–H'*). The increase in Sima protein levels in response to infection did not corroborate with a similar increase in *sima mRNA* levels at 6HPI, when only a mild increase in *sima mRNA* levels was noticed (*Figure 3—figure supplement 1I*). These indicated additional translational or post-translational control of Sima protein expression in blood cells upon wasp-infections.

Genetic perturbation of Sima expression in blood-progenitor cells by expressing *sima^RNAi*, severely impaired lamellocyte formation (*Figure 3C,D and T,U*, *Figure 3—figure supplement 1J,K and L*). The knock-down efficiency of this RNAi line in the lymph gland was confirmed by staining with Sima antibody. A significant downregulation of Sima protein expression in *domeMESO>sima^RNAi*-expressing progenitor cells was seen (*Figure 3—figure supplement 1M–N' and Q*). Similar to *Gat* and *Ssadh* knock-down, loss of Sima function did not result in dramatic defects in cell densities (*Figure 3—figure supplement 1L* and *Supplementary file 2*). Like *Gat^RNAi* and *Ssadh^RNAi* conditions, progenitor analysis revealed a marginal increase in Dome⁺Pxn⁺ intermediate progenitor cells but not overall differentiation (*Figure 3—figure supplement 1R–T* and *Supplementary file 3* and *5*).

Sima transcriptionally controls the upregulation of *lactate dehydrogenase* (*Ldh*) (*Lavista-Llanos et al., 2002*). Ldh is a key enzyme of the glycolytic pathway and in conditions of infection its upregulation has been implicated in metabolically reprogramming immune cells for an efficient cellular immune response (*Bajgar et al., 2015*; *Dolezal et al., 2019*; *Krejčová et al., 2019*). We observed a 30-fold increase in *Ldh mRNA* expression in lymph glands post-wasp-infection (*Figure 3—figure supplement 2A*). Downregulating *Ldh* expression in progenitor cells was sufficient to mimic lamellocyte reduction and indicated a requirement in the lamellocyte differentiation process as well (*Figure 3—figure supplement 2B–C', D and F*).

## GABA catabolism establishes lamellocyte potential by regulating Sima protein stability in blood-progenitor cells

Based on these findings, Sima protein levels in *Gat* and *Ssadh* loss-of-function conditions was investigated. This was undertaken by staining *Gat^RNAi* and *Ssadh^RNAi* mutant lymph glands with anti-Sima protein antibody in uninfected and wasp-infected scenarios. Compared to uninfected control 3rd instar lymph glands (*Figure 3A*), Sima protein levels in *dome-MESO>Gat^RNAi* (*Figure 3F*) and *dome-MESO>Ssadh^RNAi* (*Figure 3H*) conditions was significantly reduced (*Figure 3J*). These mutants also demonstrated a failure to raise Sima protein levels post-infection (*Figure 3K,M and O*). Succinate

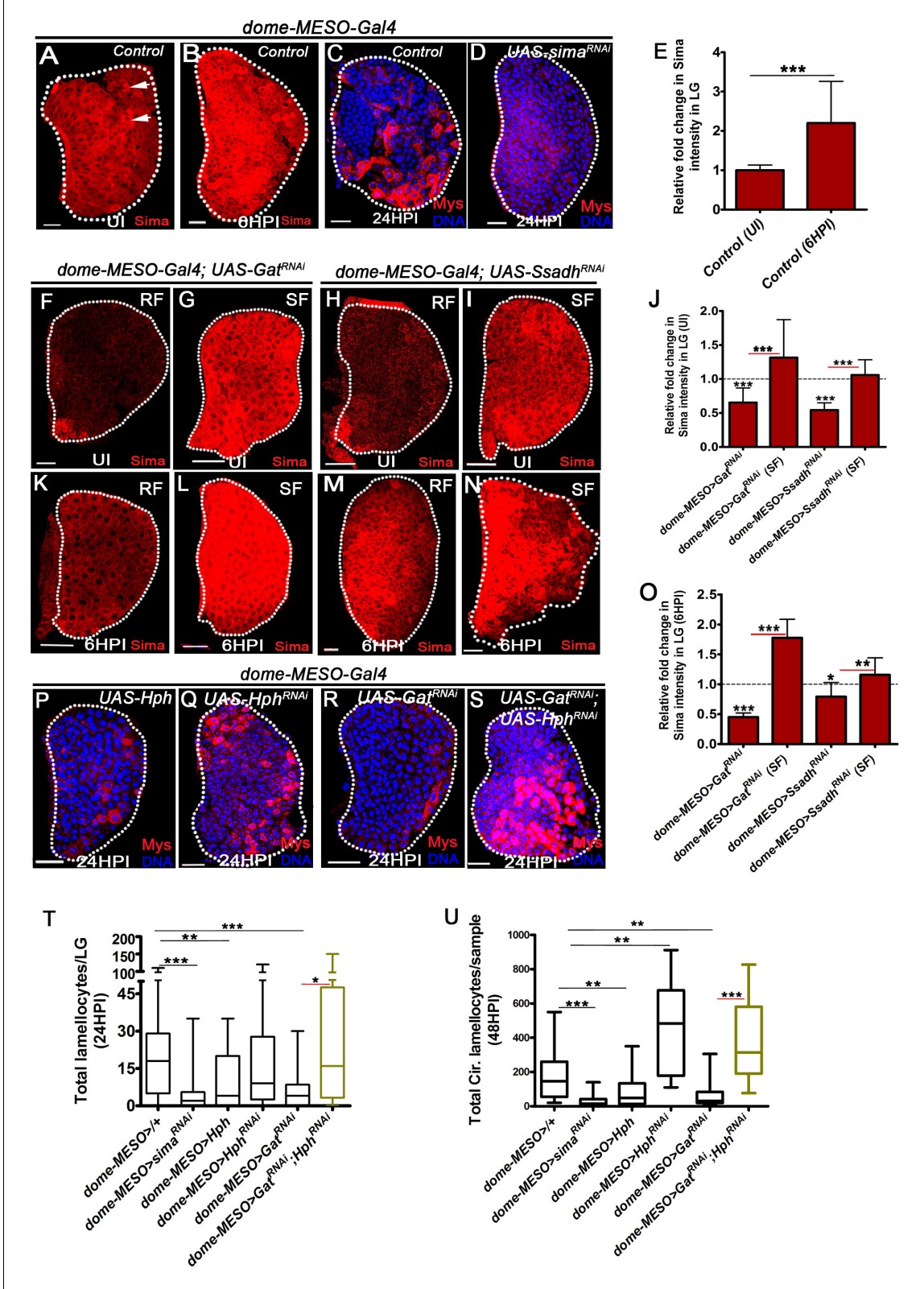

**Figure 3.** GABA-shunt-dependent control of Sima protein stabilization in immune-progenitor cells promotes lamellocyte induction. DNA is marked with DAPI (blue). In panels (C, D, P–S), Myospheroid (Mys) antibody staining (red) is employed to mark the lamellocytes in lymph gland. In panels (A-D, F-I, K-N and P-S) scale bars = 20 μm. UI indicates uninfected, HPI indicates *hours post-wasp-infection*. In lymph gland, lamellocytes analyzed at 24HPI and in circulation at 48HPI. In panel (E, J, O) mean with standard deviation is shown and in panels (T and U) median is shown in the box plots and vertical

*Figure 3 continued on next page*

*Figure 3 continued*

bars represent upper and lowest cell counts. Statistical analysis applied in panel (**E, J, O**), is unpaired *t*-test, two-tailed and in panel, (**T and U**), is Mann-Whitney test, two-tailed. 'n' is the total number of larvae analyzed, and for lymph gland 'n' represents lymph gland lobes analyzed. White dotted lines demarcate lymph glands. For better representation of the lymph gland primary lobes, the images shown, have been edited for removal of adjacent tissues (like dorsal vessel and ring gland). (**A–B**) Sima expression detected in control (*dome-MESO-Gal4, UAS-GFP/+*) lymph gland obtained from (**A**) uninfected animals (crystal cells marked with white arrows), (**B**) Sima expression is elevated at 6HPI. See corresponding quantifications in (**E**). (**C, D**) Wasp-infection response in (**C**) control (*dome-MESO-Gal4, UAS-GFP/+*) showing lamellocytes in lymph gland, (**D**) Expressing *sima*$^{RNAi}$ (*dome-MESO-Gal4, UAS-GFP; UAS-sima*$^{RNAi}$) in progenitor cells causes reduction in lamellocyte numbers in lymph gland. See corresponding quantifications in (**T**). (**E**) Relative fold change in Sima expression in control lymph glands (*dome-MESO-Gal4, UAS-GFP/+*) from uninfected and infected states at 6HPI. Compared to uninfected control lymph gland (*dome-MESO>GFP/+*, n = 12), almost twofold increase in Sima expression is observed at 6HPI (*dome-MESO>GFP/+*, n = 16, \*\*\*p=0.0007). (**F–I**) Compared to Sima levels detected in developing uninfected lymph glands from (**A**) control (*dome-MESO-Gal4, UAS-GFP/+*), (**F**) *dome-MESO-Gal4, UAS-GFP; UAS-Gat*$^{RNAi}$ show reduction in Sima expression which gets elevated (**G**) when supplemented succinate in food (SF). Similarly, (**H**) Sima expression in *dome-MESO-Gal4, UAS-GFP; UAS-Ssadh*$^{RNAi}$ also show reduction, (**I**) which gets elevated on succinate food (SF). See corresponding quantifications in (**J**). (**J**) Relative fold change in Sima expression in uninfected lymph glands control (*dome-MESO-Gal4, UAS-GFP/+*, n = 13), *dome-MESO-Gal4, UAS-GFP; UAS-Gat*$^{RNAi}$ on RF (n = 23, \*\*\*p<0.0001) and on succinate food, SF (*dome-MESO-Gal4, UAS-GFP; UAS-Gat*$^{RNAi}$, SF, n = 13, \*\*\*p<0.0001). Similarly compared to control (*dome-MESO-Gal4, UAS-GFP/+*, n = 9), *dome-MESO-Gal4, UAS-GFP; UAS-Ssadh*$^{RNAi}$ on RF (n = 11, \*\*\*p<0.0001) and on SF, (*dome-MESO-Gal4, UAS-GFP; UAS-Ssadh*$^{RNAi}$, SF, n = 8, \*\*\*p<0.0001). (**K–N**) Compared to Sima level elevation seen in (**B**) control at 6HPI, (**K**) *dome-MESO-Gal4, UAS-GFP; UAS-Gat*$^{RNAi}$ failed to show the elevation at 6HPI; however, it gets restored on (**L**) succinate food (SF). Similarly, (**M**) *dome-MESO-Gal4, UAS-GFP; UAS-Ssadh*$^{RNAi}$ failed to show the elevation at 6HPI which gets restored on (**N**) succinate food (SF). See corresponding quantifications in (**O**). (**O**) Relative fold change in Sima expression in lymph glands at 6HPI, compared to control on RF (*dome-MESO-Gal4, UAS-GFP/+*, n = 11), *dome-MESO-Gal4, UAS-GFP; UAS-Gat*$^{RNAi}$ on RF (n = 11, \*\*\*p<0.0001), and on succinate food, *dome-MESO-Gal4, UAS-GFP; UAS-Gat*$^{RNAi}$ on SF (n = 9, \*\*\*p<0.0001). Similarly compared to control, *dome-MESO-Gal4, UAS-GFP/+*, *dome-MESO-Gal4, UAS-GFP; UAS-Ssadh*$^{RNAi}$ on RF (n = 8, \*p=0.0316), and on SF (*dome-MESO-Gal4, UAS-GFP; UAS-Ssadh*$^{RNAi}$, SF, n = 9, \*\*p=0.0097). (**P–S**) Compared to (**C**) control, (*dome-MESO-Gal4, UAS-GFP/+*) lymph gland response, (**P**) expressing Hph (*dome-MESO-Gal4, UAS-GFP; UAS-Hph*) in progenitor cells causes reduction in lamellocyte numbers in lymph gland. However, (**Q**) expressing Hph$^{RNAi}$ (*dome-MESO-Gal4, UAS-GFP; UAS-Hph*$^{RNAi}$) leads to comparable number of lamellocytes in lymph gland, (**R**) *dome-MESO-Gal4, UAS-GFP; UAS-Gat*$^{RNAi}$ which shows reduction in lymph gland as compared to (**C**) control, (**S**) expressing Hph$^{RNAi}$ (*dome-MESO-Gal4, UAS-GFP/UAS-Gat*$^{RNAi}$; *UAS-Hph*$^{RNAi}$) in Gat$^{RNAi}$ background results into restoration of lamellocyte formation in lymph gland. See corresponding quantifications in (**T**). (**T**) Quantifications of lymph gland lamellocyte counts in *dome-MESO-Gal4, UAS-GFP/+* (control, n = 81), *dome-MESO-Gal4, UAS-GFP; UAS-sima*$^{RNAi}$ (n = 77, \*\*\*p<0.0001), *dome-MESO-Gal4, UAS-GFP; UAS-Hph* (n = 19, \*\*p=0.0046) and *dome-MESO-Gal4, UAS-GFP; UAS-Hph*$^{RNAi}$ (n = 52, ns), *dome-MESO-Gal4, UAS-GFP; UAS-Gat*$^{RNAi}$ (n = 26, \*\*\*p=0.0003) and *dome-MESO-Gal4, UAS-GFP; UAS-Gat*$^{RNAi}$; *UAS-Hph*$^{RNAi}$ (n = 16, \*p=0.0219). (**U**) Quantifications of circulating lamellocyte counts in *dome-MESO-Gal4, UAS-GFP/+* (control, n = 20), *dome-MESO-Gal4, UAS-GFP; UAS-sima*$^{RNAi}$ (n = 29, \*\*\*p<0.0001), *dome-MESO-Gal4, UAS-GFP; UAS-Hph* (n = 22, \*\*p=0.0067) and *dome-MESO-Gal4, UAS-GFP; UAS-Hph*$^{RNAi}$ (n = 18, \*\*p=0.0019), *dome-MESO-Gal4, UAS-GFP; UAS-Gat*$^{RNAi}$ (n = 24, \*\*p=0.0014) and *dome-MESO-Gal4, UAS-GFP; UAS-Gat*$^{RNAi}$; *UAS-Hph*$^{RNAi}$ (n = 9, \*\*\*p=0.0001).

The online version of this article includes the following source data and figure supplement(s) for figure 3:

**Source data 1.** Contains numerical data plotted in *Figure 3E,J,O,T and U*.
**Figure supplement 1.** Sima function during larval hematopoiesis establishes lamellocyte potential.
**Figure supplement 1—source data 1.** Contains numerical data plotted in *Figure 3—figure supplement 1I,J,K,L,Q and T*.
**Figure supplement 2.** Ldh function in immune cells necessary for lamellocyte induction.
**Figure supplement 2—source data 1.** Contains numerical data plotted in *Figure 3—figure supplement 2A,D and E*.

supplementation of these animals revealed a dramatic recovery in Sima protein levels almost comparable to that seen in controls on regular food (*Figure 3F–O*). This data was consistent with succinate-mediated restoration of lamellocyte phenotypes in Gat$^{RNAi}$ and Ssadh$^{RNAi}$ backgrounds and implied GABA function in moderating progenitor Sima levels. Further, *Gat* over-expressing lymph glands (*dome-MESO>Gat*) with elevated intracellular GABA (*Figure 2—figure supplement 2D*), when stained for Sima protein revealed elevated expression (*Figure 3—figure supplement 1O,O' and Q*). Taken together, these data showed an important requirement for intracellular GABA-catabolism in regulating Sima protein expression in lymph gland blood-progenitor cells. These data are also suggestive of GABA-breakdown into succinate whose availability moderated Sima protein levels.

Sima protein is marked for proteasomal-mediated degradation by hydroxy-prolyl hydroxylase (Hph [*Schofield and Ratcliffe, 2005*]) whose enzymatic activity is inhibited by succinate (*Mills and O'Neill, 2014*; *Pappalardo et al., 1992*; *Tannahill et al., 2013*). Hence, GABA-breakdown can moderate progenitor Sima protein levels by inhibiting Hph function through regulating succinate availability. This was tested by conducting genetic perturbations to modulate *Hph* expression in blood-progenitor cells. Over-expression of *Hph* that would downregulate Sima protein stability,

recapitulated Sima loss-of function phenotype (*Figure 3P,T and U*). The lamellocyte numbers were significantly downregulated in the lymph gland (*Figure 3P* and *Figure 3—figure supplement 1J*) and in circulation (*Figure 3U* and *Figure 3—figure supplement 1K and L*). Conversely, downregulating *Hph* function by expressing *Hph^RNAi* in progenitor cells as the means to increase Sima protein, led to a concomitant increase in lamellocyte numbers (*Figure 3Q,T and U*). In comparison to the numbers detected at 24HPI in the lymph glands (*Figure 3Q*), the extent of increase was more evident at 48HPI in circulation (*Figure 3U*). Finally, expressing *Hph^RNAi* in blood progenitor-cells lacking *Gat* expression (*dome-MESO>UAS-Gat^RNAi; UAS-Hph^RNAi*) rescued the *Gat^RNAi* lamellocyte defect significantly, both in the lymph gland (*Figure 3R–T*) and circulation (*Figure 3U*). These results confirmed a role for Hph function in progenitor cells in moderating lamellocyte development. They also confirmed an epistatic relationship between Gat and Hph function in blood-progenitor cells. Overexpressing Sima in progenitor cells using *dome-MESO>* or *Tep1V>* led to larval lethality, which hindered the epistatic relationship between Gat and Sima in progenitor cells. The lethality seen with Sima over-expression with these drivers may be a consequence of non-autnomous expression in tissues other than blood that compromised viability. However, an alternative approach where wild-type larvae were rasied on diets supplemented with additional GABA or succinate, showed elevated Sima protein expression in lymph gland blood-progenitor cells as compared to regular dietary states (*Figure 4—figure supplement 1A–D*). In response to wasp-infection, diet-supplemented animals mounted a superior lamellocyte response than seen in regular dietary condition (*Figure 4—figure supplement 1E*). The immune benefit of GABA and succinate supplementation was lost with progenitor-cell-specific (Dome$^+$) abrogation of *sima* expression (*Figure 4—figure supplement 1E*). These data implicated Sima function in Dome$^+$ blood-progenitor cells, downstream of GABA and succinate, in mediating the lamellocyte response. The data also showed that systemic GABA levels were limiting and when supplemented, animals were capable of mounting a superior lamellocyte response.

## Olfaction systemically controls blood-progenitor Sima stabilization

Animals with olfactory dysfunction (*orco^1/orco^1*, *Orco>Hid, rpr*) or abrogated for neuronal GABA synthesis (*Kurs6>Gad1^RNAi*) have reduced levels of systemic hemolymph GABA (*Shim et al., 2013*). We therefore hypothesized that in these animals the reduced systemic GABA levels could explain the loss of lamellocyte formation. Hence, animals with olfactory dysfunction and *Kurs6>Gad1^RNAi* were raised on a diet supplemented with GABA and succinate which successfully rescued lamellocyte numbers almost comparable to controls raised on regular diet. This was evident both in the lymph gland (*Figure 4A–D and F–H*, *Figure 4—figure supplement 1F*) and circulating lamellocyte counts (*Figure 4E and I* and *Figure 4—figure supplement 1G*). More importantly, the expression of Sima protein in lymph glands from these genetic conditions was also reduced (*Figure 4J,K and M* and *Figure 4—figure supplement 1H,I*), which was restored back to control levels in succinate supplemented diet (*Figure 4L,N* and *Figure 4—figure supplement 1J and K*).

Taken together, the data thus far reveal a critical dependence of the larval hematopoietic system on olfactory stimulation for GABA production, which controls blood-progenitor Sima levels. During hematopoiesis, systemic GABA-uptake and catabolism in lymph gland blood-progenitor cells inhibits Hph activity. This likley facilitates stabilization of Sima protein in progenitor cells and controls their lamellocyte potential. In response to wasp-infection, immune-progenitor cells increase their Gat expression, thereby increasing GABA uptake. This further increases progenitor Sima levels necessary for lamellocyte differentiation. Olfactory dysfunction or *Gat* and *Ssadh* mutants fail to achieve the threshold levels of Sima in blood-progenitor cells. Hence in these conditions, the animals fail to induce lamellocytes.

## Pathogenic odors induce immune priming

The establishment of lamellocyte potential by a long-range metabolic cross-talk set up by odor detection was puzzling. We therefore asked if the olfactory axis was involved in sensing wasps and if prior pathogenic odor experience during development influenced any aspect of the immune response. This was addressed by experiments mimicking *Drosophila* larvae rearing in wasp-infested scenarios as in the wild, where the chances of infection are higher. We reared *Drosophila* larvae from early embryonic stages in a food medium that was infused with wasp odors (a condition referred to

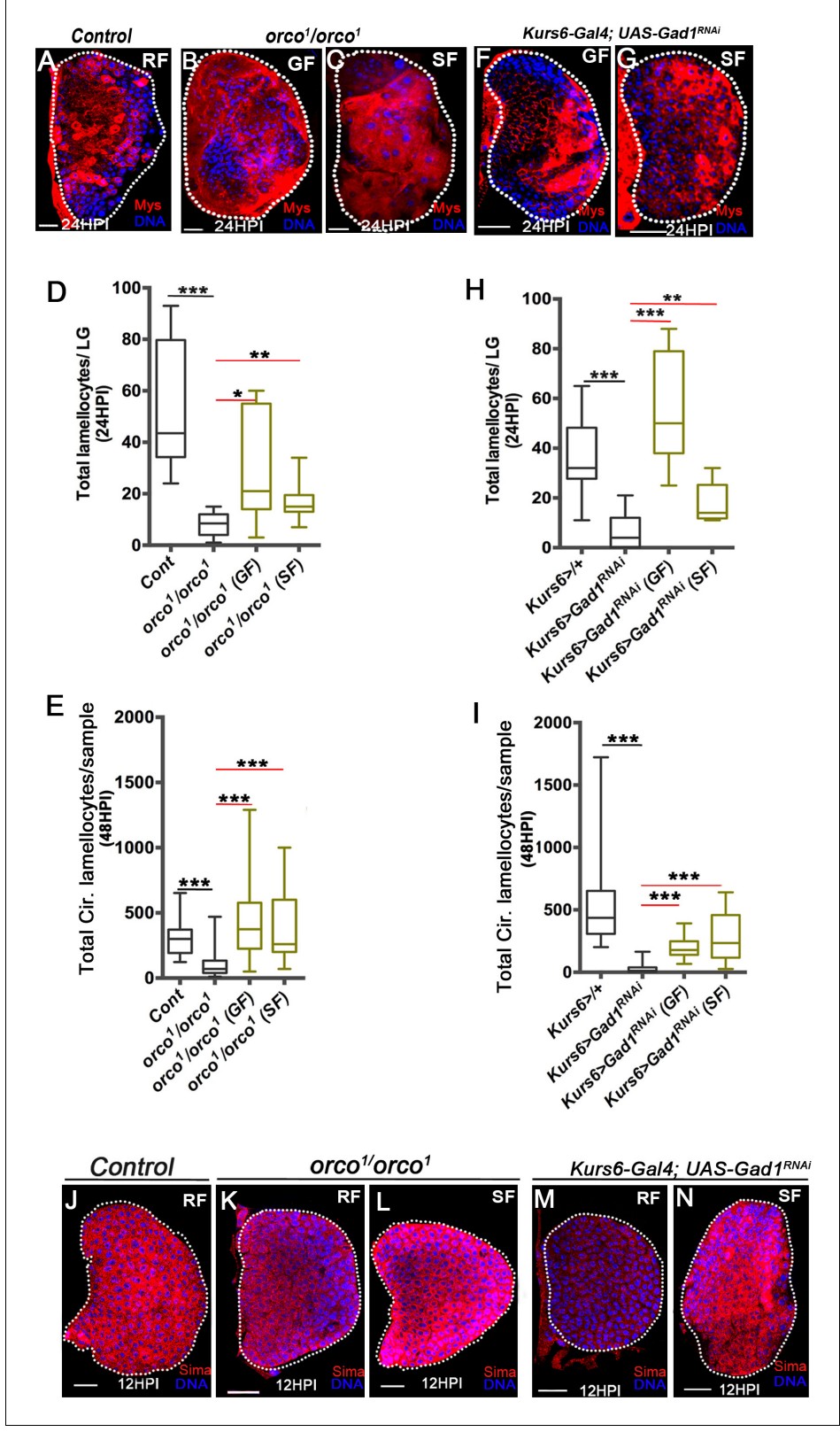

**Figure 4.** Olfaction-derived GABA and its metabolism to succinate controls lamellocyte potential. DNA is marked with DAPI (blue). Lamellocytes in panels (**A-C, F** and **G**) are marked with Myospheroid (Mys, red). Lamellocytes are characterized by their large flattened morphology. Scale bars in panels (**A-C, F, G** and **J-N** = 20 μm). HPI indicates *hours post wasp-infection*, RF is regular food, GF is GABA supplemented food, SF is succinate supplemented

*Figure 4 continued on next page*

*Figure 4 continued*

food. In lymph gland, lamellocytes analyzed at 24HPI and in circulation at 48HPI. In panels (**D, E, H** and **I**) median is shown in the box plots and vertical bars represent upper and lowest cell counts. Mann-Whitney test, two-tailed is applied for statistical analysis. 'n' represents the total number of larvae analyzed, and for lymph glands 'n' represents lymph gland lobes analyzed. White dotted lines demarcate lymph glands. For better representation of the lymph gland primary lobes, the images shown, have been edited for removal of adjacent tissues (like dorsal vessel and ring gland). (**A–C**) In response to wasp-infection, lamellocytes in lymph gland (**A**) control (*Kurs6-Gal4/+*) in regular food (RF), (**B**) *orco¹/orco¹* rescue in GABA supplemented food (GF) and (**C**) *orco¹/orco¹* rescue in succinate supplemented food (SF). (**D**) Total lamellocyte count in lymph gland from control, (*w¹¹¹⁸*, n = 8), *orco¹/orco¹* mutant in RF (n = 10, \*\*\*p<0.0001), GF (n = 7, \*p=0.022) and SF (n = 13, \*\*p=0.006). (**E**) Total lamellocyte count in circulation from control (*w¹¹¹⁸*, n = 14), *orco¹/orco¹* mutant in RF (n = 41, \*\*\*p<0.0001), GF (n = 18, \*\*\*p<0.0001) and SF (n = 14, \*\*\*p<0.0001). (**F–G**) In response to wasp-infection, lamellocytes in lymph gland, *Kurs6-Gal4, UAS-Gad1^RNAi^* in (**F**) GABA supplemented food (GF) and (**G**) succinate supplemented food (SF), compared to (**A**) control (*Kurs6-Gal4/+*) in regular food (RF). (**H**) Total lamellocytes counts in lymph gland from control (*Kurs6-Gal4/+*, n = 18), *Kurs6-Gal4, UAS-Gad1^RNAi^* in RF (n = 22, \*\*\*p<0.0001), GF (n = 11, \*\*\*p<0.0001) and SF (n = 6, \*\*p=0.007). (**I**) Total lamellocyte count in circulation in *Kurs6-Gal4/+*, (n = 25), *Kurs6-Gal4, UAS-Gad1^RNAi^* on RF (n = 25, \*\*\*p<0.0001), GF (n = 12, \*\*\*p<0.0001) and SF (n = 26, \*\*\*p<0.0001). (**J–N**) Compared to Sima protein levels detected at 12HPI in lymph glands from (**J**) control (*w¹¹¹⁸*) animals on RF, (**K**) *orco¹/orco¹* on RF and (**M**) *Kurs6-Gal4, UAS-Gad1^RNAi^* on RF show reduced Sima expression, which is restored with succinate supplementation, (**L**) *orco¹/orco¹* on SF and (**N**) *Kurs6-Gal4, UAS-Gad1^RNAi^* on SF.

The online version of this article includes the following source data and figure supplement(s) for figure 4:

**Source data 1.** Contains numerical data plotted in *Figure 4D,E,H and I*.

**Figure supplement 1.** Olfaction/GABA axis controls lamellocyte induction via modulating blood cell succinate and Sima levels.

**Figure supplement 1—source data 1.** Contains numerical data plotted in *Figure 4—figure supplement 1D,E,F, G and K*.

---

as wasp-odor food [WOF] and see Materials and methods for experimental details). These preconditioned animals were subjected to wasp immune challenge with *L. boulardi* followed by analysis of their cellular immune response. Immune response in larvae reared on regular food medium were used as experimental controls. WOF animals demonstrated a significant increase in lamellocyte numbers in response to *L. boulardi* infection (*Figure 5A,B* and *Figure 5—figure supplement 1A*). A two-fold increase at 24HPI in lymph gland lamellocyte numbers (*Figure 5A,D and E*) was evident in wasp-odor-enriched condition. Any increase in overall size of the lymph gland in WOF-infected animals was not evident (*Figure 5D,E*). An increase in circulating lamellocyte numbers at 48HPI was also evident (*Figure 5B*, *Figure 5—figure supplement 1A*), without significant difference in cell densities (*Figure 5—figure supplement 1B*). This implied that WOF condition led to more lamellocyte formation in response to wasp-infection. We also analyzed lymph glands and circulating immune cells for lamellocyte formation in homeostatic conditions. We observed that these preconditioned animals even in the absence of infection could ectopically generate lamellocytes, both in the lymph gland and circulation (*Supplementary file 1* and *2*). This recapitulated phenotypes seen with GF or SF supplemented food and also with Gat-overexpressing animals. Hence, we investigated GABA levels in animals in the homeostatic uninfected condition. Surprisingly, a twofold increase in hemolymph GABA levels was detected (*Figure 5C*). Correspondingly, lymph gland blood-cell iGABA (*Figure 5F, G*) and Sima protein (*Figure 5H,I*) levels were increased as well. This showed that WOF preconditioned animals had developmentally elevated systemic GABA availability, which consequently raised progenitor Sima expression and led to improved lamellocyte potential. The immune-benefit thus seemed unlikely of any dramatic increase in lymph gland progenitor population. Moreover, differentiation into plasmatocytes (*Supplementary file 3*) or crystal cells (*Supplementary file 5*) was comparable to controls. This implied specificity of wasp-odors in priming lamellocyte potential as opposed to controlling general blood-cell differentiation.

The increased in lamellocyte response and elevated GABA levels seen in WOF condition was detected in animals with different genetic backgrounds (*Figure 5—figure supplement 1A,C* and *Supplementary file 4*). Secondly, in *orco* mutant animals the WOF immune benefit was abrogated (*Figure 5—figure supplement 1A*). These data showed that the WOF-induced immune priming was not restricted to specific genetic backgrounds and secondly, it was mediated by olfactory

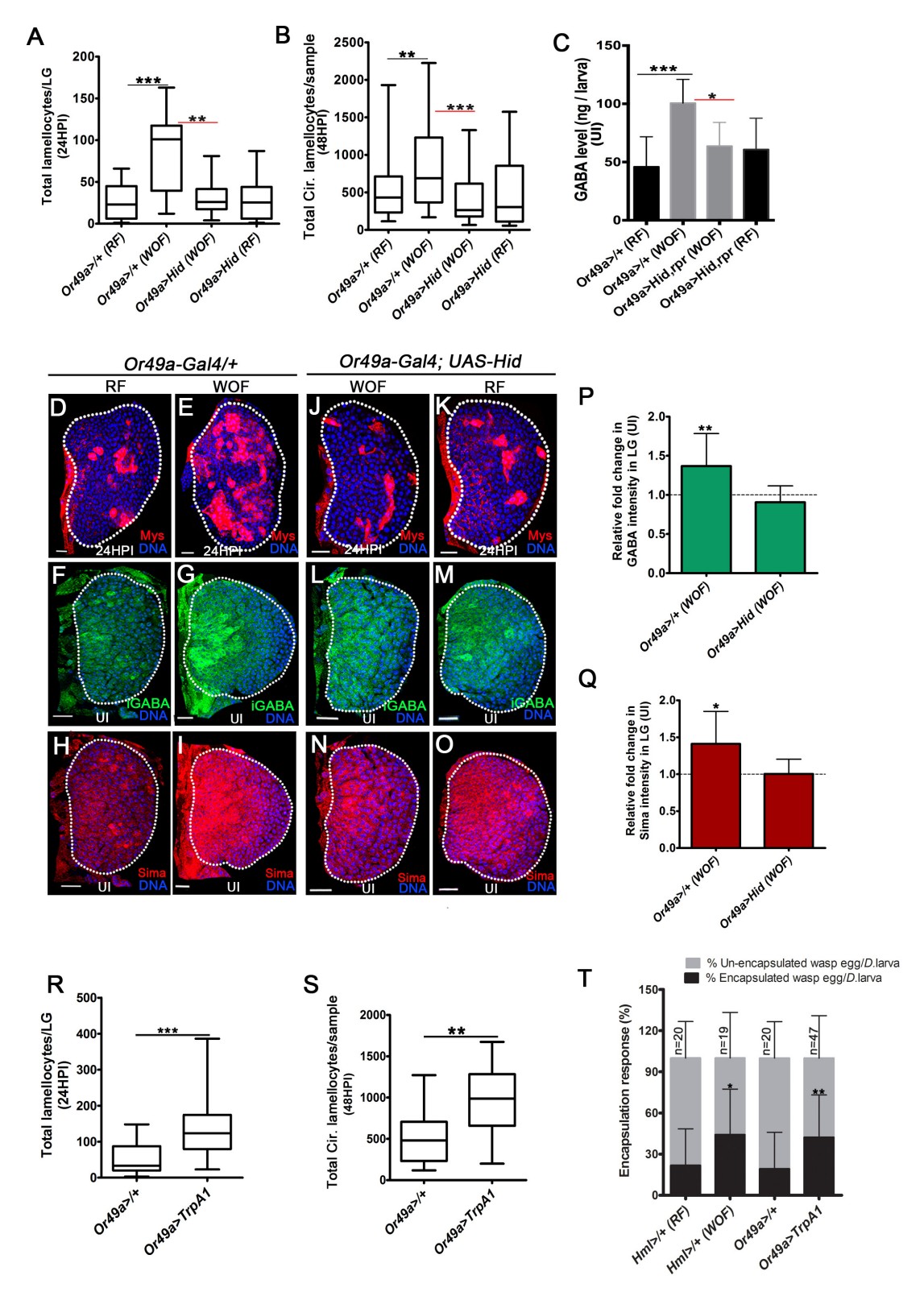

**Figure 5.** Physiological role for odors in blood cell immunity. DNA is marked with DAPI (blue), iGABA (green), Sima protein (red), and Myospheroid (Mys) staining (red) is employed to mark the lamellocytes in lymph gland Scale bars in panels D-O = 20 µm. RF is regular food, WOF is wasp-odor food and HPI indicates *hours post wasp-infection*. In lymph gland, lamellocytes analyzed at 24HPI and in circulation at 48HPI. In A, B, R and S median is shown as box plots and vertical bars represent the upper and lowest cell-counts. In panels C, P, Q, and T mean with standard deviation is shown.
*Figure 5 continued on next page*

Figure 5 continued

Statistical analysis, in panels **A, B, R, and S** is Mann-Whitney test, two-tailed and in panels **C, P, Q, and T** is unpaired *t*-test, two-tailed. 'n' represents total number of larvae analyzed, and for lymph gland 'n' represents lymph gland lobes analyzed. ns is nonsignificant. White dotted lines demarcate lymph glands. For better representation of the lymph gland primary lobes, the images shown, have been edited for removal of adjacent tissues (like dorsal vessel and ring gland). (**A**) Quantifications of lymph gland lamellocyte numbers. Compared to controls (*Or49a-Gal4/+*) raised on RF (n = 23), animal raised on WOF show increase in lamellocytes number (n = 17, ***p<0.0001). However, such increase not observed in *Or49a-Gal4, UAS-Hid* larvae when raised on WOF (n = 13, **p=0.0015) compared to *Or49a-Gal4, UAS-Hid* raised on RF (n = 20). (**B**) Quantifications of total circulating lamellocyte numbers per larvae. Compared to control (*Or49a-Gal4/+*) raised on RF (n = 38), animal raised on WOF show increase in lamellocytes number (n = 35, **p=0.0085); however, such increase was not observed in *Or49a-Gal4, UAS-Hid* larvae when raised on WOF (n = 29, ***p=0.0003) compared to *Or49a-Gal4, UAS-Hid* raised on RF (n = 29). (**C**) Quantifications of hemolymph GABA from uninfected 3rd instar larvae on RF (*Or49a-Gal4/+*, n = 27), WOF (*Or49a-Gal4/+*, n = 30 ***p=0.0008), *Or49a-Gal4, UAS-Hid, rpr* on WOF (n = 30 *p=0.011) and *Or49a-Gal4, UAS-Hid, rpr* on RF (n = 27). Refer **Supplementary file 4** for absolute amounts. (**D–E**) Compared to (**D**) control (*Or49a-Gal4/+*) lymph gland lamellocyte response, (**E**) WOF (*Or49a-Gal4/+*) animals show increased number of lamellocytes. (**F–G**) Compared to (**F**) control (*Or49a-Gal4/+*) lymph gland iGABA levels, (**G**) WOF (*Or49a-Gal4/+*) animals show elevated levels of iGABA. (**H–I**) Compared to (**H**) control (*Or49a-Gal4/+*) lymph gland Sima protein expression, (**I**) WOF (*Or49a-Gal4/+*) animals show increased Sima expression. (**J–K**) WOF condition failed to increase the lamellocyte numbers in (**J**) WOF (*Or49a-Gal4; UAS-Hid*), when compared to (**K**) RF (*Or49a-Gal4; UAS-Hid*) animals. (**L–O**) iGABA levels and Sima expression are also comparable in (**L, N**) WOF (*Or49a-Gal4; UAS-Hid*), when compared to (**M, O**) RF (*Or49a-Gal4; UAS-Hid*) animals. See corresponding quantifications in (**P and Q**). (**P**) Relative fold change in iGABA intensity of uninfected *Or49a>/+* on WOF (n = 14, **p=0.008), and *Or49a>Hid* on WOF (n = 9, ns) in comparison to RF (*Or49a>/+*, n = 11), and RF (*Or49a>Hid*, n = 5), respectively. (**Q**) Relative fold change in Sima protein intensity of uninfected *Or49a>/+* on WOF (n = 14, *p=0.011), and *Or49a>Hid* on WOF (n = 9, ns) in comparison to RF (*Or49a>/+*, n = 11) and RF (*Or49a>Hid*, n = 5), respectively. (**R**) Quantifications of lymph gland lamellocyte numbers. Compared to control (*Or49a-Gal4/+*, n = 41), forced activation of odorant receptor neuron (ORN), *Or49a* (*Or49a-Gal4; UAS-TrpA1* n = 41, ***p<0.0001) show increase in lamellocytes numbers in the lymph gland. (**S**) Quantifications of total circulating lamellocytes. Compared to control (*Or49a-Gal4/+*, n = 19), increase in lamellocytes number is seen upon forced activation of *Or49a-Gal4; UAS-TrpA1* (n = 17, **p=0.001). (**T**) Quantification of encapsulation response. Compared to control on RF (*Hml>/+*, n = 20), increased encapsulation response (%) is seen in WOF animals (*Hml>/+*, n = 19, *p=0.0256), a similar increase is also seen upon forced activation of ORN, *Or49a* (*Or49a-Gal4; UAS-TrpA1*, n = 47, **p=0.0068), as compared to control (*Or49a-Gal4>/+*, n = 20).

The online version of this article includes the following source data and figure supplement(s) for figure 5:

**Source data 1.** Contains numerical data plotted in *Figure 5A,B,C,P,Q,R,S and T*.
**Figure supplement 1.** Physiological control of cellular immunity by pathogenic wasp-odors.
**Figure supplement 1—source data 1.** Contains numerical data plotted in *Figure 5—figure supplement 1A,B,C,D,E,J,K and L*.
**Figure supplement 2.** Specific activation of projection neurons (PNs) by wasp odors.
**Figure supplement 2—source data 1.** Contains numerical data plotted in *Figure 5—figure supplement 2A,B,C,D and G*.

stimulation and not mediated by feeding or ingestion of wasp-odor components. *Drosophila* larvae exposed to other odorants (like acetic acid, 1-octen-3-ol and acetophenone [see Materials and methods for details and concentrations tested]) did not expand lamellocyte numbers or showed any increase in peripheral GABA levels in the hemolymph or in the lymph gland (*Figure 5—figure supplement 1D–J* and *Supplementary file 4*). Rather, immune cell response and GABA levels were varying in different odor conditions. This implied specificty in wasp-odors of *L. boulardi* on priming lamellocyte benefit. The data also highlighted differential control of odors on systemic GABA levels and the immune-response.

The detection of wasp odors in larvae is facilitated by activation of Or49a (*Ebrahim et al., 2015*). The ablation of Or49a (*Or49a>Hid*), diminished the immune benefits imposed by WOF condition. *Or49a>Hid* animals raised in WOF condition failed to increase their lamellocyte numbers (*Figure 5A,B and J*), hemolymph GABA level (*Figure 5C* and *Supplementary file 4*), lymph gland iGABA (*Figure 5L,P*), and Sima levels (*Figure 5N,Q*). The phenotypes detected in *Or49a>Hid* WOF animals were comparable to levels seen in *Or49a>Hid* larvae reared on regular conditions (*Figure 5A–C,K,M and O*). Importantly, loss of Or49a in regular conditions (*Or49a>Hid*, RF) did not impede the infection-induced lamellocyte response (*Figure 5A,B and K*). Neither did its loss affect hemolymph and lymph gland GABA levels in the homeostatic uninfected condition (*Figure 5C,M* and *Figure 5—figure supplement 1K*). Loss of Or49 in regular condition also did not reduce Sima protein expression in lymph glands blood cells (*Figure 5O* compared to H and *Figure 5—figure supplement 1K*). Altogether, these data showed that Or49a function was not necessary for basal lamellocyte induction or controlling developmental levels of systemic GABA levels or progenitor iGABA and Sima expression. Or49a function was however necessary for mediating the immune benefits seen in WOF condition. Genetic approaches, which force activate Or49a, recapitulated an

increase in lamellocytes as seen in WOF condition even in regular food conditions (*Figure 5R and S*). Interestingly, a similar increase was not evident with activation of Or42a (*Figure 5—figure supplement 1L*), which is unexpected as its loss abrogated lamellocyte induction. These results showed the importance of Or42a function in the establishment of basal lamellocyte potential but insufficiency in expanding their numbers. Or49a on the other hand was capable of enhancing lamellocyte potential but was dispensable for basal lamellocyte induction.

Finally, we investigated the functional implications of the increased lamellocyte phenotype on the success of the immune response. In a normal immune response, the deposited wasp-eggs are encapsulated by lamellocytes leading to the formation of a melanotic capsule and killing of the parasitoid egg. We monitored parasitoid wasp-egg encapsulation response and percent melanization response. Encapsulation response was measured by counting the number of encapsulated and unencapsulated wasp-eggs per larvae (*Vanha-Aho et al., 2015*) and for percentage melanization, infected larvae carrying melanotic wasp-egg capsules (*Yang et al., 2015*), see Materials and methods for details) post wasp-infection were estimated and represented as the percentage of larvae with black capsule to the total number of infected larvae.

In our hands, control larvae showed around 30% encapsulation response, while in WOF and *Or49a>TrpA1*, this was increased to 50% (*Figure 5T*). Furthermore, 50% control larvae showed melanization response while in WOF and *Or49a>TrpA1* condition this was also increased to 75% (*Figure 5—figure supplement 2A*). Blocking the pathway on the other hand, led to a reduction in wasp-egg encapsulation and melanization response (*Figure 5—figure supplement 2B–D*). These results highlight the physiological significance of the increased lamellocyte phenotype on effective wasp-egg clearance. Encapsulation response also requires concerted action of activated immune cells including plasmatocytes and crystal cells apart from lamellocytes (*Dudzic et al., 2015*; *Anderl et al., 2016*, *Sorrentino et al., 2002*). Therefore, an overall improved repertoire of immune cells in WOF and *Or49a >TrpA1* condition can be hypothesized. Taken together, these findings reveal the importance of environmental odor perception on cellular immune priming and function.

## Discussion

### Olfaction-immune axis in development and stress response

The olfactory-immune connect is a well-established phenomenon across systems (*Strous and Shoenfeld, 2006*). Animals with olfactory dysfunction have heightened inflammatory signatures but fail to mount immune response when challenged (*Connor et al., 2000*). The mechanistic and physiological underpinnings of olfaction and immune cross-talk in development and infection however remain poorly characterized.

In this study, we explore the importance of olfaction in cellular immune responses during *Drosophila* larval hematopoiesis. We show that as *Drosophila* larvae dwell into their food medium, sensing of food-related odors, which are the predominant odors present in their environment, leads to activation of a neuronal circuit (ORN-PN-Kurs6$^+$GABA$^+$ neuronal route). This stimulates Kurs6$^+$-GABA$^+$ neurosecretory cells to release GABA whose sensing and uptake via GABA transporter, Gat, in blood-progenitor cells of the lymph gland and intracellular breakdown establishes a non-autonomous axis that controls intracellular levels of Sima protein expression. Within 6 hr of infection with parasitic wasps, immune progenitor cells upregulate Gat protein expression. This enables cells to internalize more GABA and its metabolism further raises progenitor Sima protein expression and transcriptional activation that leads to progenitor differentiation into lamellocytes. Our data suggest a transcriptional role for Sima in promoting a metabolic shift in blood-progenitor cells which are in agreement with existing literature (*Bajgar et al., 2015*; *Dolezal et al., 2019*; *Krejčová et al., 2019*) via activation of its target gene, *Ldh* whose function is also necessary for lamellocyte formation. In the absence of food odors, or in anosmic animals, the lack of olfactory input blocks neuronal GABA production and release. This subsequently affects progenitor GABA-metabolism and Sima protein expression leading to loss of immune potential necessary for lamellocyte formation. The basal activation of the olfactory circuit through food odors is central for specifying lamellocyte cell fate. We posit that in animals dwelling in conditions where chances of infection are high, this systemic route can be co-opted to raise their immune output. The prior sensing of wasps, by *Drosophila* larvae early in development via Or49a raises downstream PN-activity leading to enhanced GABA production and

elevation of blood-progenitor cell Sima protein levels. This leads to superior immune-priming of progenitor cells and when infected these animals generate lamellocytes more rapidly and effectively (*Figure 6*).

The study reveals an unconventional developmental and stress-sensing role for the olfactory system in the sustenance of a competitive repertoire of immune progenitor cells. The utilization of Or42a, the most predominant OR activated in response to food-related odors, establishes a route that systemically connects the olfactory modality to the development of the immune system (ORN-PN-Kurs6-GABA-hematopoietic cells). The systemic connection sensitizes immune cells to environmental stressors such as the presence of wasps and promotes an innate immune training component when growing in conditions with higher chances of infection. However, it is not the overall strength of the olfactory input that controls immune-efficiency, but more specific to certain odors or activation of specific ORNs that can ultimately increase neuronal GABA production. The genetic data on forced activation of Or42a and Or49a support this notion, where activation of Or49a reciprocated with an enhancement of immune response as seen in wasp-odor enriched conditions, while activation of Or42a (*Or42a>TrpA1*) did not. This suggests that the stimulation of Or49a in the wasp-odor condtion is able to further raise downstream neuronal activity that can enhance GABA production. This raises the question of the impact of other pathogenic odors on immune-priming. Exposure of larvae to odors of varying natures (both attractive and aversive) provides some insight. Specifically, exposure to 1-octen-3-ol, did not elevate GABA levels or increase lamellocyte differentiation. 1-octen-3-ol, is a fungal aversive odorant that has been shown to affect larval plasmatocyte responses by controlling nitric oxide signaling (*Inamdar and Bennett, 2014*). This reflects the specificity of wasp-odors, its sensing and downstream signaling route employed to elevate GABA and mount lamellocyte priming.

An understanding of the complexities of cross-talk between individual ORN and their respective glomeruli and how they control PN activity is beginning to emerge and especially for specialized ORNs like Or49a, their modality of signaling is very unique (*Berck et al., 2016*). The early activation of Or49a in larval development is perceived as a stress response and is a bonus for the animal in priming immune potential. Larval Or49a is tuned to detect iridomyrmecin, which is an odor produced specifically by *Leptopilina* wasps (*Ebrahim et al., 2015*). Hence, a similar consequence on immune-priming with iridoid-producing *Leptopilina* can be predicted, but still remains to be tested.

## Dual-use of GABA in the development of a competent hematopoietic system

Within the lymph gland, the blood-progenitor cells express both GABA$_B$R and Gat. While extracellular ligand-dependent GABA/GABA$_B$R signaling promotes progenitor maintenance, the intracellular GABA metabolic pathway controls immune-differentiation potential. Ligand functions of GABA via binding to GABA$_B$R in progenitor cells elevates intracellular Ca$^{2+}$ which is necessary for their maintenance in their undifferentiated state (*Shim et al., 2013*). GABA uptake by Gat and its intracellular catabolism to promote Sima levels in blood cells on the other hand sustains a metabolic state necessary for their competency to differentiate into lamellocytes. Thus, immune cells therefore employ dual-use of GABA both as a developmental cue and as an inflammatory cue during hematopoietic development to maintain a demand-adapted immune-response. The two pathways run parallel to each other and loss of either does not impede the functioning of the other. When blood cell GABA metabolism is abrogated, blood-progenitor differentiation into lamellocyte is affected, but overall lymph gland development is unaffected and is unlike *GABA$_B$R1$^{RNAi}$* expressing blood-progenitor cells. On the other hand, when blood cell GABA$_B$R signaling is aborogated, blood-progenitor cell maintenance is affected but lamellocyte differentiation is unperturbed. Rather *GABA$_B$R1$^{RNAi}$* expressing blood progenitor cells show a mild increase in Sima protein levels (*Figure 3—figure supplement 1P,P' and Q*) alongside formation of more lamellocytes. Thus blood-progenitor cells switch from a maintenance role for GABA (GABA/ GABA$_B$R/Ca$^{2+}$ signaling) to its inflammatory function (GABA metabolism) in response to wasp-infection, which we hypothesize is at the level of Gat expression. This notion is supported by upregulation of Gat expression in response to wasp-infection, preceding the initiation of the inflammatory cellular response. Secondly, progenitor-cells overexpressing Gat even in homeostasis generate lamellocytes independent of infection. Thus limiting levels of Gat expression in progenitor-cells emerges as a potential regulator of GABA's role as an inflammatory molecule.

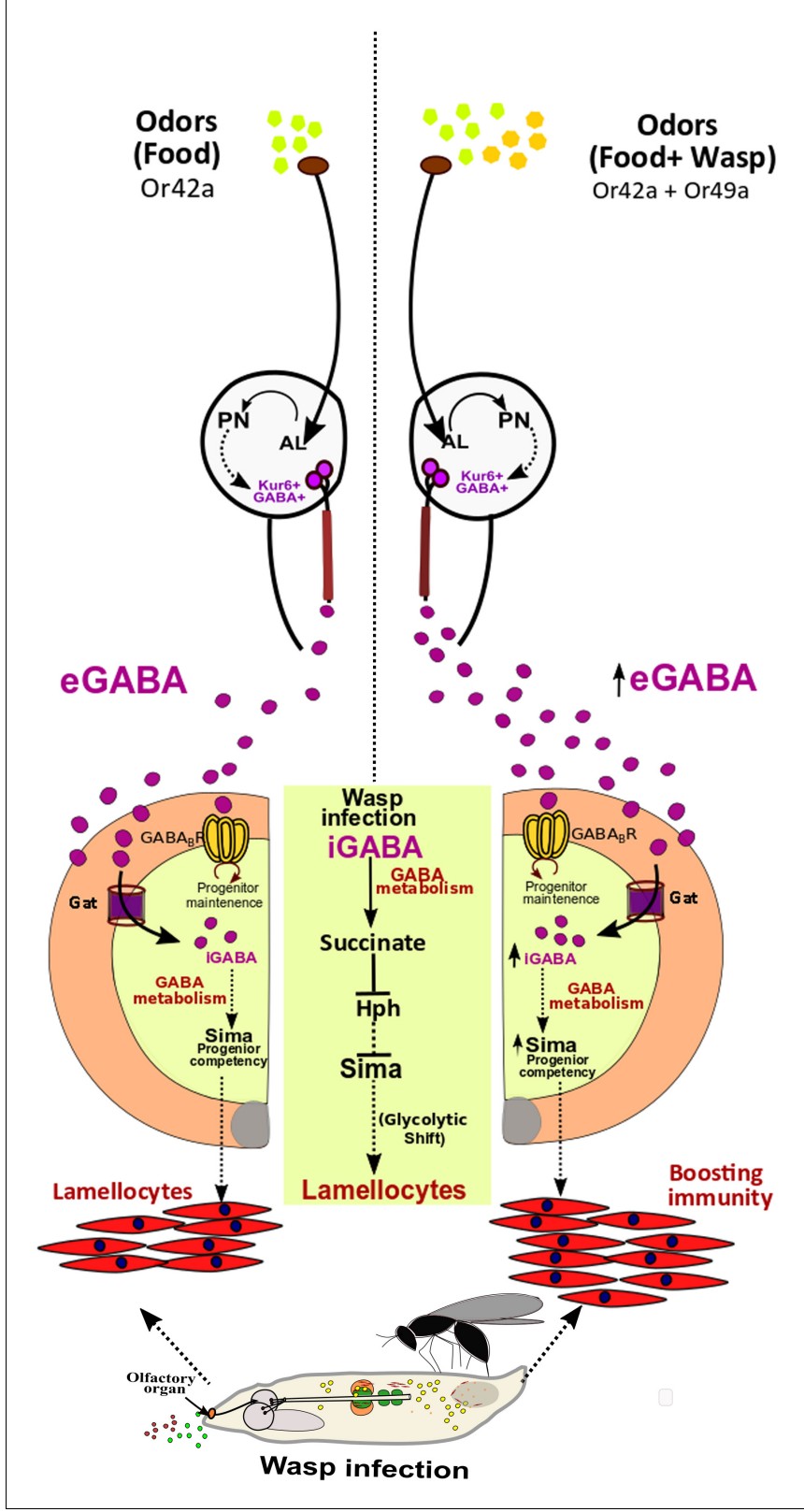

**Figure 6.** Developmental control of immune-competency by environmental odors. *Drosophila* larvae spend most of their time dwelling in food. The odors derived from this eco-system defines an integral immune-component during hematopoiesis. Sensing food odors via Or42a stimulates projection neurons (PN) leading to downstream activation of Kurs6$^+$GABA$^+$ neurosecretory cells, which mediate release of GABA (eGABA) into the hemolymph. *Figure 6 continued on next page*

*Figure 6 continued*

eGABA is internalized by lymph gland blood-progenitor cells via GABA-transporter (Gat) and its subsequent intracellular catabolism leads to stabilization of Sima protein in them. This establishes their immune-competency to differentiate into lamellocytes. Physiologically, this sensory odor axis is co-opted to detect environmental pathogenic wasp-odors. Upon detection of wasp odors via Or49a in the preconditioned media (WOF), a combinatorial stimulation of both Or42a and Or49a, elevates neuronal GABA release, leading to increase blood cell iGABA and Sima expression. This developmentally establishes superior immune-competency to withstand the immune-challenge by parasitic wasps.

The role of GABA as a general immune modulator is beginning to emerge from studies in vertebrates as well. Both GABA$_B$R and Gat are detected in immune cells of myeloid (*Stuckey et al., 2005*) and lymphoid origin (*Jin et al., 2013*). Macrophages shift to GABA as a metabolic resource to mediate inflammatory responses (*Tannahill et al., 2013*). GABA function in human hematopoietic stem cells (HSCs) or progenitor cells remains unclear and to our knowledge, any direct metabolic involvement of secreted neuronally derived GABA in hematopoietic progenitor cells has not been demonstrated. The findings from our work project commonalities between the mammalian immune system and the *Drosophila* hematopoietic system. To what extent GABA function, as we have described herein *Drosophila* through sensory routes is relevant in human HSCs or common myeloid progenitor cells remains to be investigated.

## Conclusions

Being a key pro-survival sensory modality, this study expands our current understanding of olfaction beyond modulation of animal behavior, implying more diverse physiological contexts (*Riera et al., 2017*) than previously known. Most often, animals in the wild dwell in surroundings with pathogenic threats in their environment. Such is also the case with *Drosophila* larvae in the wild where almost 80% are infected with wasps of *Leptopilina* species (*Fleury et al., 2004*). Larvae being more vulnerable to infection with their limited abilities to avoid predators in their environment, an adaptive mechanism that enhances immunity, poses a viable option to withstand such challenges. The control of inflammatory response by olfactory cues may therefore have arisen as a means to deal with unfavorable conditions. The use of general broad-odors to establish basic immune-potency that can be further modulated depending on environmental conditions, exemplifies a rheostat-like control by the olfactory axis. To our knowledge, such impact of odor-experience as a direct handle to fine-tune immune metabolism to enable an immune advantage is the first in vivo description of its kind. It will be interesting to determine if elements of the olfaction/immune axis described here are relevant for general myeloid development and adaptation. Studies such as these will lead to an understanding of how immunity is controlled by smell, as well as provide insights in the deployment of olfactory routes to train immunity in development and disease.

## Materials and methods

**Key resources table**

| Reagent type (species) or resource | Designation | Source or reference | Identifiers | Additional information |
|---|---|---|---|---|
| Genetic reagent (*D. melanogaster*) | *dome-MESO-Gal4, UAS-EYFP* | U. Banerjee | | |
| Genetic reagent (*D. melanogaster*) | *Tep4-Gal4, UAS-mCherry* | U. Banerjee | | |
| Genetic reagent (*D. melanogaster*) | *Hml△-Gal4, UAS-2xEGFP* | S.Sinenko | | |
| Genetic reagent (*D. melanogaster*) | *UAS-Gat* | M. Freeman (*Muthukumar et al., 2014*), (*Mazaud et al., 2019*) | | |

*Continued on next page*

*Continued*

| Reagent type (species) or resource | Designation | Source or reference | Identifiers | Additional information |
|---|---|---|---|---|
| Genetic reagent (*D. melanogaster*) | UAS-Hph | C.Frei | | |
| Genetic reagent (*D. melanogaster*) | Kurs6-Gal4 | G. Korge | | |
| Genetic reagent (*D. melanogaster*) | Kurs6-Gal4; mCD8GFP | Banerjee Lab | | |
| Genetic reagent (*D. melanogaster*) | Orco-gal4 | Bloomington *Drosophila* Stock Center | BL 26818 RRID:BDSC_26818 | |
| Genetic reagent (*D. melanogaster*) | Or49a-Gal4 | Bloomington *Drosophila* Stock Center | BL 9985 RRID:BDSC_9985 | |
| Genetic reagent (*D. melanogaster*) | Or42a-Gal4 | Bloomington *Drosophila* Stock Center | BL 9969 RRID:BDSC_9969 | |
| Genetic reagent (*D. melanogaster*) | GH146-Gal4 | *Shim et al., 2013* | | |
| Genetic reagent (*D. melanogaster*) | UAS-2xEGFP | Bloomington *Drosophila* Stock Center | BL 6658 RRID:BDSC_6658 | |
| Genetic reagent (*D. melanogaster*) | orco$^1$ | Bloomington *Drosophila* Stock Center (*Shim et al., 2013*) | BL 23129 RRID:BDSC_23129 | |
| Genetic reagent (*D. melanogaster*) | UAS-Hid, rpr | Nambu J.R. (*Wing et al., 1998*) | | |
| Genetic reagent (*D. melanogaster*) | UAS-Hid | Bloomington *Drosophila* Stock Center | BL65403 RRID:BDSC_65403 | |
| Genetic reagent (*D. melanogaster*) | UAS-TrpA1 | Bloomington *Drosophila* Stock Center | BL 26263 RRID:BDSC_26263 | |
| Genetic reagent (*D. melanogaster*) | TRIC | Bloomington *Drosophila* Stock Center (*Gao et al., 2015*) | BL61680 RRID:BDSC_61680 | |
| Genetic reagent (*D. melanogaster*) | : Gad1$^{RNAi}$ | Bloomington *Drosophila* Stock Center (*Shim et al., 2013*) | BL 28079 RRID:BDSC_28079 | |
| Genetic reagent (*D. melanogaster*) | ChAT$^{RNAi}$ | Bloomington *Drosophila* Stock Center (*Shim et al., 2013*) | BL25856 RRID:BDSC_25856 | |
| Genetic reagent (*D. melanogaster*) | GABA$_B$R1$^{RNAi}$ | Bloomington *Drosophila* Stock Center (*Shim et al., 2013*) | BL 28353 RRID:BDSC_28353 | |
| Genetic reagent (*D. melanogaster*) | Gat$^{RNAi}$ | Bloomington *Drosophila* Stock Center (*Stork et al., 2014*) | BL 29422 RRID:BDSC_29422 | |
| Genetic reagent (*D. melanogaster*) | Gat$^{RNAi}$ | Vienna *Drosophila* RNAi Center (*Stork et al., 2014*) | VDRC:v13359/GD FlyBase ID:FBgn0039915 | |

*Continued on next page*

*Continued*

| Reagent type (species) or resource | Designation | Source or reference | Identifiers | Additional information |
|---|---|---|---|---|
| Genetic reagent (*D. melanogaster*) | Ssadh<sup>RNAi</sup> | Vienna *Drosophila* RNAi Center | VDRC:v106637/KK FlyBase ID:FBgn0039349 | |
| Genetic reagent (*D. melanogaster*) | Ssadh<sup>RNAi</sup> | Bloomington *Drosophila* Stock Center | BL55683 RRID:BDSC_55683 | |
| Genetic reagent (*D. melanogaster*) | Ssadh<sup>RNAi</sup> | Vienna *Drosophila* RNAi Center | VDRC: v14751/GD FlyBase ID:FBgn0039349 | |
| Genetic reagent (*D. melanogaster*) | CG3379<sup>RNAi</sup> <sub>(aKDH)</sub> | Bloomington *Drosophila* Stock Center | BL 34101 RRID:BDSC_34101 | |
| Genetic reagent (*D. melanogaster*) | skap<sup>RNAi</sup> | Bloomington *Drosophila* Stock Center | BL 55168 RRID:BDSC_55168 | |
| Genetic reagent (*D. melanogaster*) | SdhA<sup>RNAi</sup> | Vienna *Drosophila* RNAi Center | VDRC:v330053 FlyBase ID:FBgn0261439 | |
| Genetic reagent (*D. melanogaster*) | sima<sup>RNAi</sup> | Bloomington *Drosophila* Stock Center (***Wang et al., 2016***) | BL33894 RRID:BDSC_33894 HMS00832 | |
| Genetic reagent (*D. melanogaster*) | Hph<sup>RNAi</sup> | Vienna *Drosophila* RNAi Center (***Mukherjee et al., 2011***) | VDRC:v103382/KK FlyBaseID: FBgn0264785 | |
| Genetic reagent (*D. melanogaster*) | Ldh<sup>RNAi</sup> | Bloomington *Drosophila* Stock Center (***Li et al., 2017***) | BL33640 RRID:BDSC_33640 | |
| Antibody | Anti-P1 (Mouse) | I. Ando | | IF(1:100) |
| Antibody | Anti-Pxn (Rabbit) | J. Shim | | IF(1:2000) |
| Antibody | Anti-PPO (Rabbit) | H. Müller | | IF(1:1000) |
| Antibody | Anti-Hnt (Mouse) | Developmental Studies Hybridoma Bank | DSHB Cat# 1g9, RRID:AB_528278 | IF(1:100) |
| Antibody | Anti-Mys (Mouse) | CF.6G11; Developmental Studies Hybridoma Bank | Cat#CF6G11 RRID:AB_528310 | IF(1:100) |
| Antibody | Anti-GABA (Rabbit) | Sigma-Aldrich | Cat# A2052 | IF(1:100) |
| Antibody | Anti-Sima (Guinea pig) | U. Banerjee | | IF(1:100) |
| Antibody | Anti-Gat (Rabbit) | M. Freeman (***Muthukumar et al., 2014***) | | IF(1:5000) |
| Antibody | Anti-pCaMKII (Rabbit) | Cell Signaling (***Shim et al., 2013***) | Cat# 3361 | IF(1:100) |
| Antibody | Anti-wingless (Mouse) | Developmental Studies Hybridoma Bank | Cat#4D4 RRID:AB_528512 | IF(1:10) |

*Continued on next page*

*Continued*

| Reagent type (species) or resource | Designation | Source or reference | Identifiers | Additional information |
|---|---|---|---|---|
| Antibody | Anti-Ci (Rat) | Developmental Studies Hybridoma Bank | Cat#2A1 RRID:AB_2109711 | IF(1:5) |
| Antibody | Anti-Antp | Developmental Studies Hybridoma Bank | Cat#8C11 RRID:AB_528083 | IF(1:100) |
| | Phalloidin | Sigma-Aldrich | Cat# 94072 | 1:100 |

## *Drosophila* husbandry, stocks, and genetics

The following *Drosophila* stocks were used in this study: $w^{1118}$ (wild type, *control*) *dome-MESO-Gal4, UAS-EYFP* and *Tep4-Gal4, UAS-mCherry* (U. Banerjee), *Hml$^{\triangle}$-Gal4, UAS-2xEGFP* (S.Sinenko), *UAS-Gat* (M. Freeman) (*Muthukumar et al., 2014*), (*Mazaud et al., 2019*), *UAS-Hph* (C.Frei), *Kurs6-Gal4* (G. Korge), *Kurs6-Gal4; mCD8GFP* (Banerjee Lab), *Orco-gal4* (BL 26818), *Or49a-Gal4* (BL 9985), *Or42a-Gal4* (BL 9969), *GH146-Gal4* (*Shim et al., 2013*), *UAS-2xEGFP* (BL 6658), *orco$^{1}$* (BL 23129 *Shim et al., 2013*), *UAS-Hid, rpr* (Nambu J.R. Wing et al., 1998), *UAS-Hid* (BL65403), *UAS-TrpA1* (BL 26263), *TRIC* (BL61680) (*Gao et al., 2015*). The *RNAi* stocks were obtained either from Vienna (VDRC) or Bloomington (BDSC) stock centres. The lines used for the study are: *Gad1$^{RNAi}$* (BL 28079 *Shim et al., 2013*), *ChAT$^{RNAi}$* (BL25856 *Shim et al., 2013*), *GABA$_B$R1$^{RNAi}$* (BL 28353 *Shim et al., 2013*), *Gat$^{RNAi}$* (BL 29422 *Stork et al., 2014*), *Gat$^{RNAi}$* (VDRC 13359/GD, *Stork et al., 2014*), *Ssadh$^{RNAi}$* (VDRC 106637/KK), *Ssadh$^{RNAi}$* (BL55683) and *Ssadh$^{RNAi}$* (14751/GD), *CG3379$^{RNAi}$* ($\alpha$KDH, BL 34101), *skap$^{RNAi}$* (BL 55168), *SdhA$^{RNAi}$* (VDRC 330053), *sima$^{RNAi}$* (HMS00832, BL33894 *Wang et al., 2016*), *Hph$^{RNAi}$* (VDRC 103382 *Mukherjee et al., 2011*), *Ldh$^{RNAi}$* (BL33640 *Li et al., 2017*), *Tgo$^{RNAi}$* (BL 26740, VDRC 10735 *Mukherjee et al., 2011*). All fly stocks were reared on corn meal agar food medium with yeast supplementation at 25°C incubator unless specified. The crosses involving RNAi lines were maintained at 29°C to maximize the efficacy of the *Gal4/UAS-RNAi* system. Controls correspond to either $w^{1118}$ (wild type) or Gal4 drivers crossed with $w^{1118}$.

All the RNAi stocks were tested for their knockdown efficiencies by using a ubiquitous driver to express these lines followed by isolation of total mRNA from whole animals subjecting them to qRT-PCR analysis with respective primers. RNAi knockdown efficiencies of the respective lines are: *Gat$^{RNAi}$* (97.7%), *CG3379$^{RNAi}$* ($\alpha$KDH, 95%), *Ssadh$^{RNAi}$* (45%), and *skap$^{RNAi}$* (40%).

All stocks were tested for their background effects for lamellocyte response to *L. boulardi* infection. This was done by crossing the respective *Gal4* lines, RNAi lines and genetic rescue combinations to $w^{1118}$ followed by wasp-infection and assessment of lamellocyte numbers (*Figure 5—figure supplement 2G*).

## Wasp infections

*Leptopilina boulardi* were maintained as previously described (*Schlenke et al., 2007*). Wasp infection protocol was followed as described in published literature (*Bajgar et al., 2015*; categorized as strong infections). Briefly, 40 *Drosophila* larvae (aged 60 ± 2 hr after egg laying) were exposed to 10 females and five male wasps for a duration of 6 hr at 25°. After removing wasps, the infected *Drosophila* larvae were put back to 29° (for RNAi crosses).

## Wasp-infection resistance assays

For encapsulation response, individual *Drosophila* larvae (60+12HPI) were sorted under stereomicroscope according to the presence or absence of black capsules. The numbers of encapsulated and un-encapsulated wasp-eggs per larvae were counted. The egg was scored as encapsulated when traces of melanin were found on it (as described in *Vanha-Aho et al., 2015*). For percent melanization, individual infected *Drosophila* larvae (60+12HPI) were sorted under stereomicroscope according to the presence or absence of black capsules. Larvae without obvious black capsules were dissected to confirm whether they were infected. The number of larvae in the cohort that showed

this melanization response was obtained as represented as the percentage of larvae with black capsule to the total number of infected larvae, as described in *Yang et al., 2015*.

## Immunostaining and immunohistochemistry

For staining circulating cells, 3rd instar larvae were collected and washed in 1X PBS and transferred to Teflon coated slides (Immuno-Cell #2015 C 30) followed by staining protocol previously described (*Jung et al., 2005*). Lymph glands isolated from larvae were also stained following similar staining protocol. Immunohistochemistry on lymph gland and circulating blood cells was performed with the following primary antibodies: mouse αP1 (1:100, I. Ando), rabbit αPxn (1:2000, J. Shim),), rabbit αPPO (1:1000, H. Müller), mouse αHnt (1:100, DSHB), mouse αMys (1:100, CF.6G11; DSHB), rabbit αGABA (1:100, Sigma, A2052), guinea pig αSima (1:100, U. Banerjee), rabbit αGat (1:5000, M. Freeman (*Muthukumar et al., 2014*), rabbit αpCaMKII (1:100, Cell Signaling, 3361 *Shim et al., 2013*), αwingless(1:10, DSHB), rat αCi (1:5, DSHB), mouse αAntp (1:100, DSHB). The following secondary antibodies were used at 1:500 dilutions: FITC, Cy3 and Cy5 (Jackson Immuno Research Laboratories and Invitrogen). Phalloidin (Sigma-Aldrich # 94072) was used at 1:100 dilutions to stain cell morphologies and nuclei were visualized using DAPI. Samples were mounted with Vectashield (Vector Laboratories). A minimum of five independent biological replicates were analyzed from which one representative image is shown.

## Lamellocyte quantification in lymph gland and circulation

Lamellocytes were identified primarily based on their large flattened morphology using cytoskeletal marker phalloidin (*Small et al., 2014*) and myospheroid/L4 (*Anderl et al., 2016*). Their quantifications were undertaken both in the lymph gland and in circulation. In uninfected conditions, both lymph gland and circulatory lamellocyte counts were done in wandering 3rd instar larvae. In wasp-infected condition, lymph gland lamellocyte response was assessed at 24 HPI prior to their disintegration and release into the hemolymph (*Lanot et al., 2001*). Circulating lamellocyte counts were done at 48 HPI, when the lamellocytes in circulation are derived from both the lymph gland pool of differentiating blood cells and circulating blood cells. For total circulating lamellocyte counts, individual larvae were bled per well and all the lamellocytes were manually counted. For lymph gland lamellocyte count, the tissues were counter-stained with phalloidin or Myospheroid and imaged to obtain Z-stacks. Large flattened lamellocytes detected by phalloidin and Myospheroid-positive cells (as lamellocytes) were manually counted per lobe of the lymph gland.

## Blood cell density analysis

To calculate circulating blood cell count/mm$^2$ and the proportion of lamellocytes in blood cells, individual larvae were bled per well, counter-stained with DAPI and phalloidin and imaged to obtain five images per well under constant magnification of 20X. All hemocytes (DAPI-positive cells) in these images were counted using the ImageJ software plugin Analyze particles tool and the numbers of cells from the five images were summed and a cell density per mm$^2$ was obtained. The respective number of lamellocytes per image was counted and plotted as proportions in blood cell count/mm$^2$. The circulating cellular response to infection was quantified at 48HPI and in uninfected conditions was undertaken in wandering 3rd instar larvae. In all experiments, control genotypes were analyzed in parallel to the experimental tests. For each experiment, a minimum of five biological replicates were analyzed and the quantifications represent the mean of all the biological replicates.

## Imaging

Immuno-stained images were acquired using Olympus FV1000 and FV3000 confocal microscopy or Nikon C2 Si-plus system under a ×20 air or ×40 oil-immersion objective. Bright field images were obtained on OlympusSZ10 or Zeiss Axiocam.

## Quantification of lymph gland phenotypes

All images were quantified using ImageJ software. Lymph gland area analysis was done as described (*Shim et al., 2012*). Roughly, middle three confocal Z-stacks were merged and threshold, selected and area was measured. This was done for respective zones and the area is represented in percent values. Controls were analyzed in parallel to the tests every time. A minimum of five animals were

analyzed each time and the experiment was repeated at least three times. The quantifications represent the mean of the three independent experimental sets. For crystal cell quantification, total number of Hnt-positive cells per lymph gland lobe was counted and represented as crystal cells per lobe. A minimum of five animals were analyzed each time and this was repeated atleast twice. For quantifying mean intensities in lymph gland tissues it was calculated as described in literature (*Louradour et al., 2017*; *Morin-Poulard et al., 2016*). Briefly, the relevant stacks of the lymph gland images were selected; the area to be measured per lobe was defined using the select tool, mean intensities were calculated in the respective selected area. Background subtractions were done by subtracting fixed squared boxes outside the lymph gland image and calculating final mean intensity. The relative fold change in intensities per lobe was calculated using mean intensity values. For intensity quantifications, the imaging settings were kept constant for each individual experimental setup. Controls were analyzed in parallel to the tests every time. A minimum of five animals were analyzed each time and the experiment was repeated at least three times.

## GABA measurements

GABA measures in circulation were conducted by bleeding five wandering 3rd instar larvae to extract their hemolymph as previously published (*Shim et al., 2013*) and analyzed using LC-MS/SRM method (Agilent 1290 Infinity UHPLC). This was done for minimum of 15 larvae per genotype and repeated three times. The quantifications shown represent the mean of all the repeats.

## In situ hybridization

Digoxigenin (DIG)-labeled probes for in situ hybridization was synthesized by PCR using DIG RNA labeling kit (Roche #11175025910). The probes for *Ssadh* and *CG33791* (α*KDH*) genes were generated using primers mentioned previously that were fused to a T7 promoter sequence. Finally the probes were applied to dissected lymph gland tissues prepared for hybridization following the previously published protocol (*Shim et al., 2012*).

## Quantitative real-time PCR analysis

Total RNA was extracted using Trizol reagent (Invitrogen, USA). For lymph gland analysis, RNA was obtained from 3rd instar larvae (#150 for each genotype). The total RNA extracted was reverse transcribed with Super Mix kit (Invitrogen) and followed by quantitative real-time PCR (qPCR) with SYBR Green PCR master mix kit (Applied Biosystems). The relative expression was normalized against *rp49* gene. The respective primers used are the following:

```
rp49 Forward: CGGATCGATATGCTAAGCTGT
rp49 Reverse: GCGCTTGTTCGATCCGTA
rps20 Forward: CTGCTGCACCCAAGGATA
rps20 Reverse: AGTCTTACGGGTGGTGAT
Gat Forward: TGCCTTGTTTCCCTACGTTC
Gat Reverse: GTACCAAGTCCAAGCCCGTA
Ssadh Forward: TTAGGAATTGCGGACAGACC
Ssadh Reverse: CTGTCCGCCCAGAATAATGT
Idh Forward: AAGCGCGTAGAGGAGTTCAA
Idh Reverse: AAGACGGTTCCTCCCAAGAT
Gdh Forward: ACGAGATGATCACCGGCTAC
Gdh Reverse: GACAGGGCTTTGACCTCATC
αKDH Forward: CGCGAATTCTCTCTCCACGCCCGCAAATC
αKDH Reverse: CGCTCTAGAGTCTCACCTGTTCCACCCTCACCA
skap Forward: CGCGAATTCGGAACCTCAATGTCCAGGAACACG
skap Reverse: GCGTCTAGAGAACTCACGGCGGGGGAACT
Sdh Forward: GTCCCACGACATTAG
Sdh Reverse: GCCAAGATAGCGGATAGC
sima Forward: AACTATCGCGAGGAGTCGAA
sima Reverse: CGTTAGCAGGGGCATATCAT
Ldh Forward: ACGGCTCCAACTTTCTGAAG
Ldh Reverse: GCAAAATGGTATCGGGACTG
```

## Minimal odor and odor-infused food preparation

The minimal odor food was prepared as described previously (*Shim et al., 2013*). WOF was prepared by placing sealed dialysis tubing with low molecular weight cut-off (Spectra/Por Dialysis tubing MWCO 500–1000 D) containing *L. boulardi* wasps in the proportion of 15 females and 5–8 males into regular food medium. This setup allows odorant cues to pass through without diffusion of any macromolecular substance. The food was freshly prepared each time. For exposure to others odorant (acetic acid, 1-octen-3-ol and acetophenone), larvae were treated with respective odorants of the highest available purity (>99% from Sigma). Acetic acid (Sigma 2722), 1-octen-3-ol (Sigma O5284) and acetophenone (Sigma 00790) at concentrations as described previously (*Kreher et al., 2005*). Briefly, the respective odorants were constituted in Mineral oil (Sigma M5904) to obtain $10^{-2}$ dilution. Of diluted odorant, 40 µl was placed on Whatman filter paper, which was then placed inside the vial containing regular fly-food media, not in direct contact with the food. Every 18–24 hr, 40 µl of odorant was again infused into the same Whatman paper to provide constant exposure of odors into the vial.

*Drosophila* larvae from 1st instar stage were reared in the different odorant infused food media until wandering 3rd instar stage. To quantify the influence of odors in immune response, each experimental set was undertaken with a minimum of 10 larvae analyzed in each set and this was repeated a minimum of three times for every odor condition.

## Succinate and GABA supplementation

Succinate (Sodium succinate dibasic hexahydrate, Sigma S2378) and GABA (Sigma, A2129) enriched diets were prepared by supplementing regular fly food with respective amounts by weight/volume measures of succinate or GABA to achieve 3% or 5% concentrations. Eggs/first instar larvae were transferred in these supplemented diets and reared until analysis of respective tissues.

## Statistical analyses

All statistical analyses were performed using GraphPad Prism six software and Microsoft Excel 2010. The medians were analyzed using Mann-Whitney test, two-tailed and means were analyzed with unpaired *t*-test, two-tailed. Images were processed utilizing ImageJ (NIH) and Adobe Photoshop CS5.

## Acknowledgements

We thank U Banerjee for Sima antibody, M Freeman for Gat antibody, Shannon Olsson for odor experiments, N Mortimer and T Schlenke for *L.boulardi* stock and FlyBase, VDRC (Austria), and BDSC for fly stocks, NCBS, CCAMP for their Fly facility, imaging, and metabolomics facilities. We thank Apurva Sarin and inStem colleagues for helpful discussion and comments on the manuscript. We specially acknowledge Varadharajan Sundaramurthy and Neeraja Subhash for imaging support in times with campus restrictions due to Covid-19 crisis. Due to space limitations, we apologize to our colleagues whose work is not cited. This study was supported by the DBT-Center of Excellence grant BT/PR13446/COE/34/30/2015, DST-ECR ECR/2015/000390, 000390DBT-IYBA 2017, CEFIPRA and DBT Ramalingaswami Re-entry Fellowship to TM and Basic Science Research Program through National Research Foundation (NRF-2014S1A2A2028388 and NRF-2017R1C1B2007343) to JS. SM is a Graduate Student at inStem, in the Tina Mukherjee lab.

## Additional information

### Funding

| Funder | Grant reference number | Author |
|---|---|---|
| Department of Biotechnology , Ministry of Science and Technology | DBT/PR13446/COE/34/30/2015 | Tina Mukherjee |
| Department of Science and Technology, Ministry of Science and Technology | DST/ECR/2015/000390 | Tina Mukherjee |

| | | |
|---|---|---|
| Department of Biotechnology , Ministry of Science and Technology | Ramalingaswami Fellowship | Tina Mukherjee |
| DBT-The Innovative Young Biotechnologist Award (IYBA) 2017 | 000390DBT-IYBA 2017 | Tina Mukherjee |
| Centre Franco-Indien pour la Promotion de la Recherche Avancée | CEFIPRA | Tina Mukherjee |
| National Research Foundation | NRF2014S1A2A2028388 | Jiwon Shim |
| National Research Foundation | NRF-2017R1C1B2007343 | Jiwon Shim |

The funders had no role in study design, data collection and interpretation, or the decision to submit the work for publication.

## Author contributions

Sukanya Madhwal, Conceptualization, Resources, Data curation, Formal analysis, Validation, Investigation, Visualization, Methodology, Writing - original draft, Writing - review and editing; Mingyu Shin, Ankita Kapoor, Manish K Joshi, Pirzada Mujeeb Ur Rehman, Resources, Data curation, Formal analysis, Validation, Investigation, Visualization, Methodology; Manisha Goyal, Resources, Data curation, Validation, Investigation, Visualization, Methodology, Writing - review and editing; Kavan Gor, Data curation; Jiwon Shim, Supervision, Funding acquisition, Investigation, Project administration; Tina Mukherjee, Conceptualization, Resources, Data curation, Software, Formal analysis, Supervision, Funding acquisition, Validation, Investigation, Visualization, Methodology, Writing - original draft, Project administration, Writing - review and editing

## Author ORCIDs

Sukanya Madhwal (ID) https://orcid.org/0000-0002-9818-7576
Jiwon Shim (ID) http://orcid.org/0000-0003-2409-1130
Tina Mukherjee (ID) https://orcid.org/0000-0003-3776-5536

## Decision letter and Author response
Decision letter https://doi.org/10.7554/eLife.60376.sa1
Author response https://doi.org/10.7554/eLife.60376.sa2

# Additional files

## Supplementary files
• Supplementary file 1. Table representing total lamellocytes counts in lymph gland tissues from uninfected wandering 3rd instar larvae.

• Supplementary file 2. Table representing blood cell counts and lamellocytes count (per $mm^2$) in circulation from uninfected wandering 3rd instar larvae.

• Supplementary file 3. Table representing lymph gland area quantifications.

• Supplementary file 4. Table representing hemolymph GABA measurements.

• Supplementary file 5. Table representing crystal cell counts in lymph gland.

• Transparent reporting form

## Data availability
All data generated or analyzed during this study are included in the manuscript and supporting files. Source data files have been provided for Figures 1–5, the corresponding Figure supplements and Supplementary files 1–5.

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
