## [Decision Letter]

**Acceptance summary:**

The authors investigate the role of the GABA-shunt pathway that generates succinate, in the lymph gland (LG) hematopoietic progenitors in response to wasp parasitism. The authors propose that wasp odor boosts the *Drosophila* immune response by regulating iGABA and Sima levels in lymph glands. This reveals an interesting cross-talk between olfaction and the immune system.

**Decision letter after peer review:**

[Editors’ note: the authors submitted for reconsideration following the decision after peer review. What follows is the decision letter after the first round of review.]

Thank you for submitting your work entitled "Control of cellular immune-competency by odors in *Drosophila*" for consideration by *eLife*. Your article has been reviewed by a Senior Editor, a Reviewing Editor, and three reviewers. The following individuals involved in review of your submission have agreed to reveal their identity: Balint Z Kacsoh (Reviewer #2); Dan Hultmark (Reviewer #3).

Our decision has been reached after consultation between the reviewers. Based on these discussions and the individual reviews below, we regret to inform you that your work has been rejected but you are allowed to submitted to *eLife* a revised version of this article that will be considered as a new submission. This allows you to have more time to address reviewers' comments. If you find that this is too risky, you have also the possibility to transfer this article to another journal.

At this stage, too many controls are lacking and the paper does not stand by itself in the absence of the other joint article. The idea would be to reinforce the conclusion by (1) centering the paper on the lymph gland, (2) by clearly demonstrating the effect of wasp odors on hematopoiesis, (3) by characterizing the pathway (GABA-Succinate-SIMU) et, (4) by revealing the consequence of resistance on wasp infection.

Reviewer #1:

In this study, the authors established that odors participate in the control of lamellocyte differentiation, a specific immune cell type which appears in *Drosophila* in response to wasp parasitism and is required for wasp egg encapsulation. They propose that odor sensing leads to the release of GABA into the hemolymph, which is taken up by lymph gland progenitors. Raising *Drosophila* larvae on food enriched in GABA (GF) or succinate (SF) rescues defects observed in those that are unable to sense odors. They further observed an increase in Sima levels in lymph gland cells in response to wasp parasitism. Finally, they propose that wasp odor contributes to an efficient *Drosophila* immune response by regulating GABA and Sima levels in lymph glands.

This analysis is interesting but must be significantly improved to support the conclusions proposed by the authors. The statements drawn in the summary, as well as in the first paragraph of the discussion, are not in agreement with the data presented. Furthermore, many conclusions rely on data that are neither published nor provided in the manuscript; this is not acceptable. Several key controls are missing (see below for details). Concerning the organization of the manuscript: in the Results section only data should be given, information relative to Materials and methods/tools or comments/discussion should be shifted to the corresponding sections. The key results of the study (including controls) must be provided in the main figures and the reader should be able to understand the conclusions by analyzing them. A strong reorganization of figures (including the addition of controls) is requested to improve data presentation for clarity and to prevent the reader from getting confused among main and supporting data.

Measuring larval melanisation is not an accurate way to evaluate the success of wasp encapsulation. Indeed, melanised larvae can give rise to living wasp larvae. A more reliable criterion will be the % of wasp egg hatching (%of wasp eggs that give rise to living wasp larvae).

Furthermore, in most experiments, the authors measured the absolute numbers of lamellocytes in circulation and rely on this criterion to estimate the efficiency of the *Drosophila* immune response. First, the % of circulating lamellocytes relative to the total amount of circulating blood cells (that can differ depending on genetic contexts) should be more accurate/meaningful than giving an absolute number. Second, since the correlation between circulating lamellocyte numbers and the efficiency of larval wasp encapsulation is under debate (see the recent paper by Leitao et al., 2019 and also data given in this study), it is very important to systematically analyse both lamellocyte numbers (in lymph gland and in circulation) and to measure the % of wasp egg encapsulation in every genetic context.

The data relative to Sima expression and function (by lof and gof in lymph gland progenitors) has to be given in the present manuscript. The functional links between GABA levels, Hph and Sima in lymph gland progenitors have to be established. The expression of these genes in different mutant contexts has to be performed and epistasis experiments must be done to establish their hierarchy.

To establish a functional link between Orco1 and Hph the authors performed rescue experiments of orco1 mutants by expressing hph RNAi under the control of the hml-Gal4 driver which is NOT expressed in lymph gland progenitors but in circulating differentiating blood cells (Figure 3G-J). These data do not support the proposed model where the function of Hph on Sima is supposed to occur in lymph gland progenitors! Rescue experiments must be done with hph RNAi under the control of lymph gland progenitor drivers.

Concerning the contribution of wasp odor to the *Drosophila* immune response: data provided are not convincing (see comments below for Figure 4) The injection of oil droplets to *Drosophila* larvae is known to induce the immune response and leads to droplet encapsulation. This represents a very interesting alternative to address the contribution of the wasp odor to the *Drosophila* immune response that should be tested here.

The model presented in Figure 5 does not summarize the data presented in this study. Those that link the cascade comprising iGABA, succinate, hph, sima and Ldh to lamellocyte differentiation are not provided here. The proposed role for wasp odor is not convincing. The cooperative effects of food and wasp odors were not analyzed.

Figure 1

What about the specificity of the Gal4 drivers that are used (Or42a, Kurs6)? Are similar results obtained in Or42A>hid and Or42A>or42RNAi (that would preserve neurons) larvae? What about lymph gland GABA levels in Kurs6>GatRNAi?

Figure 1—figure supplement 1

In Orco>Hid, rp and Or42b>hid: what about lymph gland lamellocyte numbers?

In RF, MOF and MOF+ food odor: what about lamellocyte differentiation in the lymph gland?

In Figure 1—figure supplement 1E in GH146>ChatRNAi there are only few lamellocytes in circulation but the % of melanised larvae does not seem statistically different from controls (statistics are missing). Thus, these results contradict the correlation between circulating lamelllocyte numbers and larval melanisation (see comments above), they also contradict the role of acetylcholine in the response to the wasp parasitism via its control of GABA levels. What about GABA and Sima levels in lymph glands in these experiments?

Are there any defects in lamellocyte numbers (in the lymph gland and in circulation), in wasp egg encapsulation, in lymph gland GABA and Sima levels when acetylcholine is constitutively released?

Figure 2

It must be established here that GABA receptor is not required, and that the role of GABA is mediated by the metabolic pathway to generate succinate. This must be added and presented in the main figures.

A-D controls are missing: phenotypes (lamellocyte differentiation in lymph glands and in circulation, GABA and Sima levels in lymph glands) should be analysed in the absence of wasp parasitism and when larvae are raised on GF and SF medium. 2E-H’ controls without parasitism should be provided. Is there any difference in larval development or size when they are raised on GF or SF compared to RF?

It is essential to illustrate in this Figure (i)the internalisation of GABA in lymph gland progenitors; (ii) the requirement of GABA internalisation for lamellocyte differentiation.

2A and 2C: what about wasp egg encapsulation?

In circulation (Figure 2E’,G’,F’,H’) since the red cells are considered as lamellocytes although they do not display their specific elongated shape, a marker for mature lamellocytes (L1, β−intergrin,.…) should be used.

Figure 3

A control lymph gland picture (without parasitism) must be presented, pictures B and D should be replaced since the focus seems to be different from the other pictures shown?

All the data relative to Sima expression and function in the lymph gland should be introduced here. What about Sima expression when larvae are raised on SF in the absence of parasitism? What about lamellocyte numbers (in the lymph gland and in circulation) and wasp egg encapsulation when Sima is overexpressed (gof) or in sima loss-of-function (lof) in lymph gland progenitors? Epistasis experiments between GABA and Sima in lymph gland progenitors must be performed.

Hph expression and function in lymph gland progenitors must be analysed. Recue and epistasis experiments between GABA, Hph and Sima must be performed in lymph gland progenitors to establish whether there are functional links between them. In hml>hphRNAi in Orco 1 mutants there is a strong increase in circulating lamellocyte numbers: is the total number of circulating blood cells affected?

3G-H: in hml>hph RNAI without wasp parasitism, what about lamellocyte numbers (in the lymph gland and in circulation) and lymph gland GABA and Sima expression? In the corresponding text the authors use the term "blood cells" for both lymph gland progenitors and circulating blood cells (as identified by the hml>Gal4 driver). This is very confusing since they are very distinct cell types. The adequate terms must be used for clarity.

Figure 2—figure supplement 2K: high numbers of circulating lamellocytes in Orco>Hid,rp larvae raised on GF and SF. What about the number of total circulating blood cells in these conditions, about lamellocyte numbers in lymph glands and wasp egg encapsulation in these contexts? What about lamellocyte numbers (in circulation and lymph gland), in the absence of parasitism?

Figure 4

Non-infected controls are missing: experiments without wasp infection must be run in parallel with those performed under wasp parasitism conditions. This is crucial to conclude that the phenotypes observed are due only to wasp infection. 4A: there is a huge dispersion of the values, the number of larvae analysed should be extended to reduce dispersion. Lamellocyte numbers in Or49a> hid-rp (RF) are similar (even superior) to Or49a>+ (RF) larvae, indicating that wasp odor is not required for lamellocyte production under regular wasp infection as it is performed in the lab. These data rather suggest that raising larvae in WOF and in the absence of infection, prime the lymph gland progenitors that are now more competent to rapidly differentiate into lamellocytes upon wasp parasitism. Longer exposure of larvae to WOF or to odor concentration might have a side effect on lymph gland progenitors in control larvae. This can be seen in Figure sup 2J where at the L3 stage, a significant alteration of dome+ cells indicates that the lymph gland progenitors differ between larvae raised on WOF compared to those raised on RF. What about GABA and Sima levels in L2/L3 lymph glands from control larvae raised on WOF medium? Analyzing the immune response triggered by oil injection in larvae might help to distinguish the contribution or not of wasp odor to this response.

Since the % of wasp egg encapsulation has not been examined the authors cannot conclude that wasp odor acts on the "efficiency of the immune response" as stated in the text.

4G is different from Figure 3A: Why?

4J is not in agreement with the quantification given in Figure 4L, similar remark holds for 4K and 4M.

4F: higher GABA levels are observed in the cardiac tube compared to the control 4E. Unfortunately, this raises doubts about the rigor with which the experiments were performed. To prevent this interrogation, pictures should not be a tight crop around one lymph gland lobe but a larger view including surrounding tissues (cardiac tube, pericardial cells) that would allow the reader to compare backgrounds between controls and experiments.

What is the control genotype in Figure 4C-H?

Sup Figure 3O-P: what about lamellocyte numbers in the lymph gland, wasp egg encapsulation, and lymph gland GABA and Sima levels?

Sup Figure 4

What about wasp egg encapsulation, lymph gland lamellocytes, lymph gland Sima levels when the different odors are provided to *Drosophila* larvae?

H-L': not convincing since the quality of pictures is not good enough. Why do we see such extensive green staining in J-L'? These data are not necessary.

*Reviewer #2*:

Madhwal et al., present their work, entitled, "Control of cellular immune-competency by odors in *Drosophila*." In this study, the authors investigate and identify a role for *Drosophila* larval environmental odor experience on priming cellular immune potential. Excitingly, the authors show that odor sensing is critical to production of lamellocytes in the circulating hemolymph of a *Drosophila* larva. This odor detection mediates the release of GABA from neurosecretory cells and is subsequently internalized by blood progenitor-cells. This internalization is followed by catabolization to generate succinate which stabilizes Sima (HIFα) protein, key for lamellocyte production. Remarkably, *Drosophila* larvae in odor environments mimicking parasitoid-threatened conditions raises systemic GABA and blood-progenitor Sima levels. Thus, these larvae have a primed immune response in anticipation of infection. Also, thank you to the authors for a wonderful summary Figure (Figure 5).

Collectively, this body of work represents novel and important insight into influence of environmental odor-experience on immune phenotypes. The genetic controls and experimental lines are elegantly chosen and the manuscript is written in a very clear and logical order. The rescue experiment with GABA or succinate supplementation is especially compelling. Odorants influence myeloid- metabolism and the priming of the innate-immune system, a truly remarkable finding building on the emerging field of environmental modulation of physiology.

It is my recommendation that this important manuscript be accepted pending revisions outlined below:

The authors provide an extremely important body of work. However, I have a few concerns on the genetic dissection of the phenotype that are important to be addressed:

– The role of *Leptopilina boulardi* venom may be a confounding variable. As described in Markus et al., 2005, a sterile needle wound is sufficient to trigger lamellocyte production and differentiation. While the data in Figure 1 is quite compelling, I believe it important to test via sterile needle wound the wild-type and Or42a>Hid line. Alternatively, sterile needle wound alone may not be sufficient to trigger the heightened response, but only in combination with odorant. This may be the case as the authors examine a general injury response. However, the methods do not outline what a general injury response is from, so I cannot conclude the finding. Either way, this would be important to address.

– The defects in melanization yield a second important question: Are crystal cells also negatively affected by the inability to detect odorants? Are crystal cell populations affected by wasp odor? This question should be investigated and can be easily by heating *Drosophila* larvae as described in the citation below and counting the melanin spots. If this cell type is also affected, it would provide a stronger mechanistic link between lack of melanotic activity and odor detection of either both cell types OR only lamellocytes.

– Crystal cells self-melanize when larvae are incubated at 60°C for 10 minutes.

– Williams, Ando and Hultmark, (2005).

– This point is furthered by text in the manuscript: "GABA metabolism does not control differentiation of blood cells to plasmatocytes or crystal cell lineages, implying specificity of GABA in priming lamellocyte potential."

– The odorant clearly primes the immune response of the *Drosophila* larvae. I am left wondering what is the odorant that does the priming. The Materials and methods read:

"*L. boulardi* wasps in the proportion of 15 females and 8 males into regular food medium".

– I believe it is important to the impact of the paper to ask whether the odorant detected is male wasp specific or female wasp specific (OR perhaps it is not specific?). Either way, this is an important outstanding question that should be addressed. Regardless of the answer, this will further catapult this exciting finding into becoming a seminal work in the field of environmental modulation of physiology. This will also provide a baseline to identify what exactly Or49a is detecting (male, female, or general wasp odor?). Pure male populations can be acquired by using virgin female wasps to infect larvae. All F1 wasps will be male, thus providing a pure odorant. I am excited to read future studies that will hopefully identify the molecule that is being detected.

*Reviewer #3*:This is a very interesting paper, throwing more light on the mysterious connection between olfaction and immunity, previously described by some of the authors of this manuscript. The data presented here show that olfactory detection of parasites, via one or two specific odorant receptors, is required to prime the immune system for an enhanced response to later parasite attacks. They also confirm that the signal from the central nervous system to the hematopoietic tissue is mediated by GABA. GABA is taken up by the blood cell precursors, affecting their cellular metabolism and stabilizing Sima, a homolog of hypoxia-inducible factor α (HIFalpha). There, Sima is required for the generation of immune response effector cells (lamellocytes). When the authors blocked olfactory signaling, for instance by mutating a key olfactory co-receptor, the animals were unable to make lamellocytes. This capacity could be rescued, for instance by directly providing GABA, or by genetically blocking Sima turnover, thereby increasing its concentration. The results are convincing, and the links described here between olfaction, HIF signaling and immunity should be of considerable general interest. However, the paper is not very well written and some important information is missing:

1) A very recent article by Krejčová et al., (2019) describes the role of HIF signaling in the activation of *Drosophila* blood cells during bacterial infection. That paper very nicely complements the results described here. It was perhaps published after this manuscript was submitted, but appropriate references to that article must be added.

2) References are made to a "manuscript in submission" by Madhwal et al. Depending on how close that manuscript has come to publication, it may be wise to depend less on data presented there, since these data are still hidden from the reader. The authors could probably make their points by referring to other sources (e.g the above-mentioned paper by Krejčová et al.,), or to the experiments shown in the present manuscript.

3) Specifically, it is claimed that "Sima is both necessary and sufficient for lamellocyte induction". The data presented here suggest that Sima is necessary, and data elsewhere point in the same direction, but I am not aware of any published data showing that it is sufficient for lamellocyte induction. If that claim is only supported by the other submitted manuscript, it is better to delete it. The presented model does not depend on it.

4) The experimental system is not fully described, leaving it to the reader to fill the gaps. For instance, it is not clearly stated which parasite is studied. The Introduction makes a general statement about "Leptopilina wasps", and in the later sections, the reader has to infer that "wasps" or "*L. boulardi* wasps" refers to wasps of the species *Leptopilina boulardi*. That may seem self-evident for people in the field, but to help others it should be stated explicitly.

5) Many figures show the effects of genetic constructs, food etc. on lamellocyte production. From the context it can be inferred that these effects were often (maybe always?) studied in wasp-infected larvae. That must be clearly stated.

6) The second sentence in the Results section introduces "a subset of neurosecretory cells (Kurs6+)", implying that Kurs6+ is a term for a specific set of neurons. Later, "Kurs6" comes up as part of a genotype. That confused me at first. It took me some time to figure out that Kurs6-Gal4 is in fact a driver construct, and that "Kurs6+" simply refers to cells that express this driver. It would have been helpful to properly introduce this driver to the reader.

7) The term "iGABA" turned up rather unexpectedly in the text, and it confused me a lot. I don't think it is understood by the general reader. Does it simply refer to intracellular GABA? If so, I strongly suggest to spell it out, rather than introducing yet another multiple-letter combination. That would not make the text significantly longer.

[Editors’ note: further revisions were suggested prior to acceptance, as described below.]

Thank you for submitting your article "Metabolic control of cellular immune-competency by odors in *Drosophila*" for consideration by *eLife*. Your article has been reviewed by Anna Akhmanova as the Senior Editor, a Reviewing Editor, and three reviewers. The following individuals involved in review of your submission have agreed to reveal their identity: Tomas Dolezal (Reviewer #1); Balint Z Kacsoh (Reviewer #2).

The reviewers have discussed the reviews with one another and the Reviewing Editor has drafted this decision to help you prepare a revised submission.

Summary:

The reviewers agree that the paper has been improved and is now easier to read. The findings were judged fascinating but there are still issues. The authors delineate a linear story (one pathway) but some elements could affect the system independently. The reviewers agree on a set of recommendations that should be addressed during the revision of the manuscript.

Essential revisions:

1) Resistance to parasitoid wasp

The authors provide an extremely important body of work. However, but the reviewers have a concern about the physiological significance of the phenotype. It is appropriate to hypothesize that an increase in lamellocyte production will yield a more potent immune response against parasitoids, as seen in other *Drosophila* species (i.e. D. suzukii). However, genetic perturbation that increase lamellocyte numbers, or perturbs the immune system in any manner, does not necessarily mean that the immune response mounted will be successful. The authors should provide experiments monitoring resistance to parasitoid wasps when the pathway they discovered is perturbated. There should monitor the impact of feeding larvae on WOF on resistance and how disturbing Or49A, Gat and Ssadh affect resistance to parasitoid wasp.

2) RNAi effectivity and using one line

The reviewers questioned the validity of the study as some results are based only one RNAi and their knockdown efficiencies were tested by using a ubiquitous and not in the actual tissues. They however recognize that the model is supported by the fact that they are testing different players affect the pathway. The reviewers however ask to repeat the experiments with Gat and Ssadh using another RNAi line to reinforce their conclusion.

3) Sima staining

Figure 3: There are discrepancies in the Sima staining which put question into the specificity of this staining/back ground. For example, some LGs showed a punctate expression of Sima in the posterior part of the LG (Figure 3F,G and H which is not seem in the other LGs). Pictures in Figure 3B, K and m are not in agreement with quantifications in 3O. The same comment holds for Figure 3F-L and quantifications in J. Expression of Sima in lamellocyte is also not convincing. The specificity of the Sima antibody has to be checked. Figure 3—figure supplement 1I is the difference in sima mRNA levels significant? The reviewers recommend to address this point or at least to prepare a supplementary figure showing replicated of the picture they use of their graph.

[Editors’ note: further revisions were suggested prior to acceptance, as described below.]

Thank you for submitting your article "Metabolic control of cellular immune-competency by odors in *Drosophila*" for consideration by *eLife*. Your article has been reviewed by Anna Akhmanova as the Senior Editor, a Reviewing Editor, and one reviewer. The reviewers have opted to remain anonymous.

The reviewers have discussed the reviews with one another and the Reviewing Editor has drafted this decision to help you prepare a revised submission.

Summary:

The authors investigate the role of the GABA-shunt pathway that generates succinate, in the lymph gland (LG) hematopoietic progenitors in response to wasp parasitism. The authors propose that wasp odor boosts the *Drosophila* immune response by regulating iGABA and Sima levels in lymph glands. This reveals an interesting cross-talk between olfaction and the immune system.

There is an interest from the reviewers to publish this manuscript in *eLife* but some issues remained. At this stage, we encourage you to address either experimentally or by changes in the text the comments listed below (a possibility is to tune down the statements). The hope is that you can rapidly submit a revised version that addresses the few points left.

Essential revisions:

1) The regulatory cascade that goes from iGAbA to Ldh (Figure 6, left part) is not fully established, since several epistasis experiments are still lacking. For example, functional links in the lymph gland are not established between: (i) hph and sima, and more importantly between (ii) sima and Ldh and (iii) sima and iGABA. Epistasis experiments are lacking which precludes drawing in the model Figure 6 plain arrows representing functional connections. Some experiments given in the manuscript for establishing functional links are irrelevant: this is the case of the epistasis experiment between oroc1 and Sima (via hph RNAi with the hml driver; sup Figure 4L-O). The hml-gal4 driver is not expressed at all in lymph gland progenitors where hph function is supposed to be required!

One key point concerns Sima functions. In the model, Sima is acting downstream of iGABA and is required for lamellocyte differentiation in response to wasp parasitism (Figure 6, left part). Unfortunately, these regulations are not yet definitively established in the manuscript.

Why have authors not performed rescue experiments of DomeMESO>Gat RNAi or (Dome-MESO>Ssadh RNAi) of the mutant lymph gland phenotype by overexpressing Sima with the Dome-MESO gal4 driver? This is a key experiment that would establish whether Sima is the key payer downstream of iGaba.

Concerning the functional link between Sima and lamellocyte differentiation: Does the overexpression of Sima with the dome-MESO gal4 driver lead to a cellular immune response similar to the one observed in response to wasp infection? These are key questions that have to be addressed to sustain the model proposed in Figure 6.

2) Figure 6 (right panel) this representation is misleading and is not in agreement with the presented data. The proposed role for wasp odor for the immune response is not correct. Indeed, killing the Or49a neurons (wasp odor sensing neurons) has no impact on the immune response on larvae raised on RF (Figure 5). Thus, there is no need at all for these neurons to mount an immune response. Raising larvae on WOF leads only to an increase in the immune response that is dependent on Or49A neurons. Any conditions that lead to increased GABA levels (such as SF, GF,WOF ) and even in the absence of wasp parasitism (since lamellocytes are detected in these conditions in the absence of parasitism) have the same consequence; i. e. a boost of the immune response. This data indicates that increasing GABA levels (by activating GABAnergic neurons) by different ways leads in all conditions to boosting the immune response.

3) Ca^2+^ levels in the lymph gland and their potential contribution to the immune response is unclear. What about Ca 2+ levels and requirement: (i) in response to wasp infection and (ii) in dome-MESO>gat RNAi conditions?

4) Figure 2—figure supplement 4 and Figure 4D-E, H-I controls are missing. This concerns Figure 4D and E; Figure 4H and I and Figure 2—figure supplement 4L. In response to wasp parasitism, do wild type larvae raised in SF or GF have increased lamellocyte differentiation compared to wt parasited larvae raised on RF?

5) Figure 2—figure supplement 3: why are the authors looking at pCamKII? What does this marker indicate?

6) Figure 2—figure supplement 3P: DomeMESO>gatRNAi , a decrease in iGABA is observed (Figure 2H) whereas no difference for pxn is measured. This is different from data given in Shim et al., 2013 where a decrease in GABA leads to increase Pxn? How can one reconcile these data?

7) Dome-MESO>sima RNAi: Subsection “Progenitor Sima protein stability via GABA-catabolism establishes lamellocyte Potential”: one cannot write that "plasmatocyte number is not affected" since quantifications are missing.

8) Propositions for deleting some parts:

– Figure 3—figure supplement 1M-T' are redundant with Figure 3.

– Data relative to tango: They are based on only one RNAi treatment and they are not essential. Thus, they could be removed.

– Figure 4A', B', C', F' and G' are not informative. Cells shown have a round shape and do not correspond to lamellocytes that have a characteristic elongated morphology.

– Figure 5—figure supplement 2G there is no reference of this Figure in the manuscript!

9) I have still comments on the writing as the article, which is difficult to follow even to quite close experts.

– The Abstract is difficult to understand:

Here is a possible (suggested) Abstract based on your texts:

“Studies in different animal model systems have revealed the impact of odors on immune cells, However, any understanding on why and how odors control cellular immunity remained unclear. We find that *Drosophila* employ an olfactory/immune cross-talk to tune a specific cell type, the lamellocytes, from hematopoietic-progenitor cells. We show that neuronally released GABA derived upon olfactory stimulation, is utilized by blood-progenitor cells as a metabolite and through its catabolism, these cells stabilize Sima/HIFα protein in them. In blood-progenitor cells, Sima capacitates these cells with the ability to drive a metabolic state that is necessary for initiating lamellocyte differentiation. This systemic axis becomes relevant for larvae dwelling in wasp-infested environments where chances of infection are higher. By co-opting the olfactory route, the preconditioned animals elevate their systemic GABA levels leading to the up-regulation of blood-progenitor cell Sima expression. This elevates their immune-potential and primes them to respond rapidly when infected with parasitic wasps. The present work highlights the importance of the olfaction in immunity and shows how odor detection during animal development is utilized to establish a long-range axis in the control of immune-progenitor competency and priming.”

Introduction “innate competitiveness” consider rewriting this term.

Introduction: The third paragraph does not fit well the flow of the text. Include this text after mentioning the link between odor and immunity.

Introduction: consider shortened this part or move some part in the Abstract.

Introduction

Indicates that you are using Orco and provide background information or replace by an introductory sentence if the result is described below.

Indicate that it is in the “lymph gland”.

Replace mutational analysis by “the use of mutations” or “the use of loss of function mutations”.

Subsection “Olfaction controls cellular immune response necessary to combat parasitic wasp Infections”: explain what is PSC and why you look at PSC.

Subsection “Progenitor Sima protein stability via GABA-catabolism establishes lamellocyte Potential”: Avoid to use a passive sentence form. It will be simpler to say. We next explored…. (this applies to many other parts of the text).

Consider to add additional sub-sectioning in the results to facilitate the reading.

Subsection “Pathogenic odors induce immune priming”, second paragraph: I could not understand the statements. Could you make it clearer?

The Discussion is far too long and should be divided by two.

---

## [Author Response]

[Editors’ note: the authors resubmitted a revised version of the paper for consideration. What follows is the authors’ response to the first round of review.]

Reviewer #1:In this study, the authors established that odors participate in the control of lamellocyte differentiation, a specific immune cell type which appears in *Drosophila* in response to wasp parasitism and is required for wasp egg encapsulation. They propose that odor sensing leads to the release of GABA into the hemolymph, which is taken up by lymph gland progenitors. Raising *Drosophila* larvae on food enriched in GABA (GF) or succinate (SF) rescues defects observed in those that are unable to sense odors. They further observed an increase in Sima levels in lymph gland cells in response to wasp parasitism. Finally, they propose that wasp odor contributes to an efficient *Drosophila* immune response by regulating GABA and Sima levels in lymph glands.This analysis is interesting but must be significantly improved to support the conclusions proposed by the authors. The statements drawn in the summary, as well as in the first paragraph of the discussion, are not in agreement with the data presented. Furthermore, many conclusions rely on data that are neither published nor provided in the manuscript; this is not acceptable. Several key controls are missing (see below for details). Concerning the organization of the manuscript: in the Results section only data should be given, information relative to Materials and methods/tools or comments/discussion should be shifted to the corresponding sections. The key results of the study (including controls) must be provided in the main figures and the reader should be able to understand the conclusions by analyzing them. A strong reorganization of figures (including the addition of controls) is requested to improve data presentation for clarity and to prevent the reader from getting confused among main and supporting data.Measuring larval melanisation is not an accurate way to evaluate the success of wasp encapsulation. Indeed, melanised larvae can give rise to living wasp larvae. A more reliable criterion will be the % of wasp egg hatching (%of wasp eggs that give rise to living wasp larvae). Furthermore, in most experiments, the authors measured the absolute numbers of lamellocytes in circulation and rely on this criterion to estimate the efficiency of the *Drosophila* immune response.

The main finding of the manuscript is the importance of sensory stimulation to generate a cell type that is necessary to respond to infections by parasitic wasps. The melanization data in the previous version was added to support the functional aspect of these immune cells and that in conditions when these cells were not detected the melanization of wasp-eggs was also undetectable. The aspect of melanization itself is far more complex and we agree with the reviewer that our assays do not evaluate it and also is not the central point of our investigation. We feel if any systemic control of melanization exists, will be better understood by undertaking a more thorough analysis and best done as an independent study. Hence, the current manuscript focuses on mechanisms controlling the development of immune-competent blood-progenitor cells that are capacitated to differentiate into lamellocytes. These are are impressive cells as they are only seen in conditions of infection/inflammation and not seen in homeostasis, unlike plasmatocytes and crystal cells. The processes that define such capability to the immune system forms the central focus of this manuscript.

First, the % of circulating lamellocytes relative to the total amount of circulating blood cells (that can differ depending on genetic contexts) should be more accurate/meaningful than giving an absolute number.

This is done. In addition to total lamellocytes numbers/larva, their respective proportions to total circulating blood cells is also provided for every genetic context in infected and uninfected conditions. The data in infected conditions are shown in: Figure 1—figure supplement 1E, Figure 2—figure supplement 1M, Figure 2—figure supplement 2C, Figure 2—figure supplement 5C, Figure 3—figure supplement 1L, Figure 5—figure supplement 1B and uninfected states is provided in Supplementary file 2.

Second, since the correlation between circulating lamellocyte numbers and the efficiency of larval wasp encapsulation is under debate (see the recent paper by Leitao et al., 2019 and also data given in this study), it is very important to systematically analyse both lamellocyte numbers (in lymph gland and in circulation) and to measure the % of wasp egg encapsulation in every genetic context.

Lymph gland lamellocyte counts are now provided for all genetic conditions. They are shown in Figure 1C, 1F, 1I, Figure 2P, Figure 3T, Figure 4D, 4H, Figure 5A and Extended 7N, 8N Figure 1—figure supplement 1A, Figure 1—figure supplement 1C, Figure 2—figure supplement 1K, Figure 2—figure supplement 1N, Figure 2—figure supplement 1D, Figure 2—figure supplement 1I, Figure 2—figure supplement 4I, Figure 2—figure supplement 5A, Figure 3—figure supplement 1J, Figure 4—figure supplement 1F.

The data relative to Sima expression and function (by lof and gof in lymph gland progenitors) has to be given in the present manuscript. The functional links between GABA levels, Hph and Sima in lymph gland progenitors have to be established. The expression of these genes in different mutant contexts has to be performed and epistasis experiments must be done to establish their hierarchy.

The functional link between GABA levels, Hph and Sima in lymph gland progenitors is provided. Figure 2, Figure 3 and Figure 1—figure supplement 1, Figure 2—figure supplement 2, Figure 2—figure supplement 3, Figure 2—figure supplement 4, Figure 2—figure supplement 5, Figure 3—figure supplement 1 and Figure 3—figure supplement 2 are completely dedicated towards describing this process.

To establish a functional link between Orco1 and Hph the authors performed rescue experiments of orco1 mutants by expressing hph RNAi under the control of the hml-Gal4 driver which is NOT expressed in lymph gland progenitors but in circulating differentiating blood cells (Figure 3G-J). These data do not support the proposed model where the function of Hph on Sima is supposed to occur in lymph gland progenitors! Rescue experiments must be done with hph RNAi under the control of lymph gland progenitor drivers.

We have attempted to do the rescue experiment as suggested by the reviewer of orco1 mutants by expressing hph RNAi under the control of a progenitor driver. However, the genetic complexity of this experiment has limited its success as the combinations were often unhealthy and failed to propagate. As a result, we have been unable to get enough larvae of the correct genotype for addressing the infection response. As we continue to attempt this question by other approaches, we utilized HmlΔGal4 to address this issue. HmlΔGal4 is a well-established lymph gland driver line. It marks differentiating blood cells of the lymph gland as early as 60h AEL ((Jung et al., 2005), (Goto et al., 2003, Mondal et al., 2011) (Charroux and Royet, 2009)). Using this driver, we find recovery of lamellocyte formation in the lymph glands of orco1 mutants. The data provides epistatic relationship of the systemic pathway in blood cells of the lymph gland and indicates that loss of Hph in Hml^+^ is also sufficient to restore their lamellocyte formation.

Concerning the contribution of wasp odor to the *Drosophila* immune response: data provided are not convincing (see comments below for Figure 4) The injection of oil droplets to *Drosophila* larvae is known to induce the immune response and leads to droplet encapsulation. This represents a very interesting alternative to address the contribution of the wasp odor to the *Drosophila* immune response that should be tested here.

We thank reviewer for this suggestion, and we will employ this assay to investigate the melanization/encapsulation response of lamellocytes in our forthcoming investigations.

The model presented in Figure 5 does not summarize the data presented in this study. Those that link the cascade comprising iGABA, succinate, hph, sima and Ldh to lamellocyte differentiation are not provided here. The proposed role for wasp odor is not convincing. The cooperative effects of food and wasp odors were not analyzed.

The model presented in Figure 5 does not summarize the data presented in this study. Those that link the cascade comprising iGABA, succinate, hph, sima and Ldh to lamellocyte differentiation are not provided here. The proposed role for wasp odor is not convincing. The cooperative effects of food and wasp odors were not analyzed.

This is corrected. The current draft incorporates all the missing data on GABA metabolism linking it to progenitor Ldh function in lamellocyte formation to strengthen the model presented.

Figure 1What about the specificity of the Gal4 drivers that are used (Or42a, Kurs6)?

The specificity of these drivers is previously shown in Shim et al., and show limited expression within the neuronal tissue without any non-specific expression in immune cells.

Are similar results obtained in Or42A>hid and Or42A>or42RNAi (that would preserve neurons) larvae?

We tested RNAi line Or42a (BL#65152) and failed to detect any difference due to its poor knock-down efficiency. There are 2 other RNAi lines available at VDRC which we were unable to procure due to difficulties with general procurement of lines from VDRC (problems with importing these lines and custom clearances that continue to exist). Unfortunately, we also tried to obtain Or42a mutant lines from Carlson lab and Dennis Mathew, but failed to obtain them for the same import reasons. We feel unfortunate, but nevertheless, I also understand the point raised by the reviewer. In response to this, I would like to mention the physiological experiment done with media lacking food odors (Figure 1—figure supplement 1C and 1D) where olfactory neurons are preserved showed defective lamellocyte formation both in the lymph gland and circulation (Figure 1—figure supplement 1C and 1D) which was corrected by restoration of food odors back into the medium. Additionally, orco1 mutant larvae also showed lamellocyte defect. Subsequently, downstream of olfactory signaling we have blocked projection neurons and Kurs6 neurons which phenocopy the lamellocyte defect. Taken together, all the data show the defect in lamellocyte formation is not a consequence of loss of Or42a, rather lack of olfactory signalling.

What about lymph gland GABA levels in Kurs6>GatRNAi?

We do not understand the concern being raised by Kurs6>GatRNAi, which would block

GABA transporter in Kurs6 neurons. However, blocking GABA-synthesis (Kurs6>Gad1RNAi) in Kurs6 neurons leads to reduction in systemic GABA levels and lymph gland GABA levels. This is previously reported in (Shim, Mukherjee et al., 2013) and we have now made this clearer in the text.

Figure 1—figure supplement 1In Orco>Hid, rp and Or42b>hid: what about lymph gland lamellocyte numbers?In RF, MOF and MOF+ food odor: what about lamellocyte differentiation in the lymph gland?

This is provided in Figure 1—figure supplement 1A-C and 1F.

Figure 2It must be established here that GABA receptor is not required, and that the role of GABA is mediated by the metabolic pathway to generate succinate. This must be added and presented in the main figures.

We provide the entire data on GABA receptor and its lack in lamellocyte formaion as an Figure 2—figure supplement 1K. To maintain the focus and narrative of the study on olfaction and GABA’s role as a metabolite and the increasing number of panels that describe this, we have kept the entire data on GABA receptor as an independent Figure 2—figure supplement 1, but will be happy to make it into a main figure if necessary.

Is there any difference in larval development or size when they are raised on GF or SF compared to RF?

Larval development or size on these diets at concentrations tested is not altered. Our data on lymph gland blood development in SF condition in Figure 2—figure supplement 4A-C, showing no change in their hematopoietic aspects reflects this and implies, that this dietary state does not induce any stress response, given the highly sensitive nature of the lymph gland immune to stresses (Shim et al., 2012) (Owusu-Ansah and Banerjee, 2009) (Kim et al., 2011).

It is essential to illustrate in this Figure (i)the internalisation of GABA in lymph gland progenitors; (ii) the requirement of GABA internalisation for lamellocyte differentiation.

Figure 2 describes GABA uptake and its internalization by lymph gland progenitors necessary for lamellocyte formation. Figure 2A-2K show dependency of Gat function in moderating intracellular GABA levels in blood progenitor cells and Figure 2L-Q describe its intracellular metabolic role in lamellocyte response. The experiments have been conducted using 2 independent blood-progenitor cell driver and these data are shown in Figure 2—figure supplement 2.

2A and 2C: what about wasp egg encapsulation?In circulation (Figure 2E’,G’,F’,H’) since the red cells are considered as lamellocytes although they do not display their specific elongated shape, a marker for mature lamellocytes (L1, β−intergrin,.…) should be used.

Lamellocytes have a characteristic large flattened morphology which is the primary phenotype that is utilized to detect and count these cells. The stainings with phalloidin or with Myospheroid (β−intergrin) have been undertaken to identify the cells of this characteristic shape. We have used protocols to detect these cells that are routinely used and published (Anderl et al., 2016), (Small et al., 2014). Panels with Myospheroid (β−intergrin) staining are now shown in Figure 2L-O, Figure 3C, 3D, 3P-S, Figure 4A-C, 4F, 4G, Figure 5D, 5E, 5J, 5K, Figure 2—figure supplement 4D-H, Figure 3—figure supplement 3B-C.

Figure 3A control lymph gland picture (without parasitism) must be presented, pictures B and D should be replaced since the focus seems to be different from the other pictures shown?All the data relative to Sima expression and function in the lymph gland should be introduced here. What about Sima expression when larvae are raised on SF in the absence of parasitism? What about lamellocyte numbers (in the lymph gland and in circulation) and wasp egg encapsulation when Sima is overexpressed (gof) or in sima loss-of-function (lof) in lymph gland progenitors? Epistasis experiments between GABA and Sima in lymph gland progenitors must be performed.Hph expression and function in lymph gland progenitors must be analysed. Recue and epistasis experiments between GABA, Hph and Sima must be performed in lymph gland progenitors to establish whether there are functional links between them. In hml>hphRNAi in Orco 1 mutants there is a strong increase in circulating lamellocyte numbers: is the total number of circulating blood cells affected?3G-H: in hml>hph RNAI without wasp parasitism, what about lamellocyte numbers (in the lymph gland and in circulation) and lymph gland GABA and Sima expression? In the corresponding text the authors use the term "blood cells" for both lymph gland progenitors and circulating blood cells (as identified by the hml>Gal4 driver). This is very confusing since they are very distinct cell types. The adequate terms must be used for clarity.

The draft is restructured significantly. Most experiments have been redone and the current version provides the epistatic relationship between GABA and Sima in lymph gland blood progenitor cells. As mentioned, these data are presented in Figure 2, Figure 3 and Figure 1—figure supplement 1, Figure 2—figure supplement 2, Figure 2—figure supplement 3, Figure 2—figure supplement 4, Figure 2—figure supplement 5, Figure 3—figure supplement 1 and Figure 3—figure supplement 2.

With regards to hml>hphRNAi in Orco 1 mutants, our attempts to conduct the rescue experiment as suggested by the reviewer of orco1 mutants by expressing hph RNAi under the control of a progenitor driver have not been successful. The genetic complexity of this experiment has limited its success and the combinations were often unhealthy and failed to propagate. As we continue to attempt this question by other approaches, we utilized HmlΔGal4 which is also a well-established lymph gland driver line. It marks differentiating blood cells of the lymph gland as early as 60h AEL. Using this driver, we find recovery of lamellocyte formation in the lymph glands of orco1 mutants. These data provide epistatic relationship of the systemic pathway in blood cells of the lymph gland and indicates that loss of Hph in Hml^+^ is also sufficient to restore their lamellocyte formation and is provided as supplemental data. This data is also supportive of our previous findings on the GABA-pathway using HmlΔGal4, and is suggestive of a lamellocyte lineage trajectory that is: dome^+^ to dome^+^Hml^+^ to Hml^+^ to Mys^+^, in the lymph glands. This is currently under investigation.

Figure 2—figure supplement 2K: high numbers of circulating lamellocytes in Orco>Hid,rp larvae raised on GF and SF. What about the number of total circulating blood cells in these conditions, about lamellocyte numbers in lymph glands and wasp egg encapsulation in these contexts? What about lamellocyte numbers (in circulation and lymph gland), in the absence of parasitism?

Lamellocyte numbers in circulation and lymph gland, in the absence of parasitism of Orco>Hid,rp larvae is provided in Supplementary file 1 and Supplementary file 2. Post-infection the total circulating blood cell numbers are provided in Figure 1—figure supplement 1E.

Figure 4Non-infected controls are missing: experiments without wasp infection must be run in parallel with those performed under wasp parasitism conditions. This is crucial to conclude that the phenotypes observed are due only to wasp infection.

The data on un-infected WOF conditions are clearly provided in Supplementary file 1 and Supplementary file 2. We detect the formation of lamellocyte even in the absence of infection while their numbers are much reduced than detected in infections, the data indicate improved competency of blood-progenitor cells to make lamellocytes in WOF condition.

Figure 4A: There is a huge dispersion of the values, the number of larvae analysed should be extended to reduce dispersion. Lamellocyte numbers in Or49a> hid-rp (RF) are similar (even superior) to Or49a>+ (RF) larvae, indicating that wasp odor is not required for lamellocyte production under regular wasp infection as it is performed in the lab. These data rather suggest that raising larvae in WOF and in the absence of infection, prime the lymph gland progenitors that are now more competent to rapidly differentiate into lamellocytes upon wasp parasitism. Longer exposure of larvae to WOF or to odor concentration might have a side effect on lymph gland progenitors in control larvae. This can be seen in Figure Sup 2J where at the L3 stage, a significant alteration of dome+ cells indicates that the lymph gland progenitors differ between larvae raised on WOF compared to those raised on RF. What about GABA and Sima levels in L2/L3 lymph glands from control larvae raised on WOF medium? Analyzing the immune response triggered by oil injection in larvae might help to distinguish the contribution or not of wasp odor to this response.

We have attempted to clarify these points. We have added more number of samples to increase “n” values for WOF condition. But the nature of the data is the same and the dispersion is still huge. We predict this is a consequence of the nature of the experiment which shown dynamic physiology of animals and how they perceive odors and respond. However, the trend to make more lamellocytes in response to wasp-odors is significantly evident. We show upregulation of GABA and Sima levels in developing L3 lymph glands from control uninfected larvae raised on WOF medium. This is shown in Figure 5G on WOF compared to 5F on RF (GABA) and 5I on WOF compared to 5H on RF (Sima). Blocking sensing of wasp-odors by abrogating Or49a (Or49>Hid) shows a failure of these animals to increase the levels of GABA and Sima as seen in WOF. The levels in Or49>Hid animals raised on WOF, remain comparable to levels seen in controls on regular food. Figure 5L compared to 5G and 5F (GABA) and 5N compared to 5I and 5H (Sima). We have also undertaken sterile wounding of WOF animals to analyze immune response to general injury (this was shown in the previous version, but now omitted). While cellular immune response to injury is comparable to controls, these animals make more lamellocytes here as well. We feel the data on un-infected WOF conditions provided in Supplementary file 1 and Supplementary file 2 where lamellocyte formation is detected even in the absence of infection proves the point on wasp-odors and their contribution in improved competency of blood progenitor cells in making lamellocytes. Hence, for the sake of simplicity and linearity in the current version of the manuscript the data on injury in WOF animals is not provided, but we will be happy to add it back.

Since the % of wasp egg encapsulation has not been examined the authors cannot conclude that wasp odor acts on the "efficiency of the immune response" as stated in the text:Figure 4G is different from Figure 3A: Why?Figure 4J is not in agreement with the quantification given in Figure 4L, similar remark holds for 4K and 4M.Figure 4F: higher GABA levels are observed in the cardiac tube compared to the control 4E. Unfortunately, this raises doubts about the rigor with which the experiments were performed. To prevent this interrogation, pictures should not be a tight crop around one lymph gland lobe but a larger view including surrounding tissues (cardiac tube, pericardial cells) that would allow the reader to compare backgrounds between controls and experiments.What is the control genotype in Figure 4C-H?

Majority of the stainings have been re-done and the images have been quantified to strengthen the point on wasp-odor sensing and its ability to raise systemic GABA levels and lamellocyte response. The detection of wasp-odors leading to increase in systemic GABA levels raises blood-progenitor GABA and Sima expression. The point raised by the reviewer on higher GABA levels being observed in the cardiac tube on WOF conditions reflects the point that in these conditions the systemic levels of GABA is up-regulated (Figure 5G compared to 5F for lymph gland data and also see 5C for hemolymph GABA data). We have also made sure that all genotypes are mentioned.

Sup Figure 3O-P: what about lamellocyte numbers in the lymph gland, wasp egg encapsulation, and lymph gland GABA and Sima levels?Figure 5—figure supplement 1What about wasp egg encapsulation, lymph gland lamellocytes, lymph gland Sima levels when the different odors are provided to *Drosophila* larvae?H-L': not convincing since the quality of pictures is not good enough. Why do we see such extensive green staining in J-L'? These data are not necessary.

The data on different odors is mainly provided to show the specificity of wasp-odors in lamellocyte formation and modulating systemic GABA levels. Hence, we show circulating lamellocyte counts and GABA levels. We have data on Sima levels as well and will be happy to provide it. We have removed the panels H-L'.

Reviewer #2:Madhwal et al., present their work, entitled, "Control of cellular immune-competency by odors in *Drosophila*." In this study, the authors investigate and identify a role for *Drosophila* larval environmental odor experience on priming cellular immune potential. Excitingly, the authors show that odor sensing is critical to production of lamellocytes in the circulating hemolymph of a *Drosophila* larva. This odor detection mediates the release of GABA from neurosecretory cells and is subsequently internalized by blood progenitor-cells. This internalization is followed by catabolization to generate succinate which stabilizes Sima (HIFα) protein, key for lamellocyte production. Remarkably, *Drosophila* larvae in odor environments mimicking parasitoid-threatened conditions raises systemic GABA and blood-progenitor Sima levels. Thus, these larvae have a primed immune response in anticipation of infection. Also, thank you to the authors for a wonderful summary Figure (Figure 5)!Collectively, this body of work represents novel and important insight into influence of environmental odor-experience on immune phenotypes. The genetic controls and experimental lines are elegantly chosen, and the manuscript is written in a very clear and logical order. The rescue experiment with GABA or succinate supplementation is especially compelling. Odorants influence myeloid- metabolism and the priming of the innate-immune system, a truly remarkable finding building on the emerging field of environmental modulation of physiology.It is my recommendation that this important manuscript be accepted pending revisions outlined below:

We thank reviewer#2 for the positive response on our work. The suggestions made have been incorporated into the revised version of the draft and critique made have been very useful in re-writing of the manuscript. Please find our response to comments and concerns raised herewith.

The authors provide an extremely important body of work. However, I have a few concerns on the genetic dissection of the phenotype that are important to be addressed:– The role of Leptopilina boulardi venom may be a confounding variable. As described in Markus et al., 2005, a sterile needle wound is sufficient to trigger lamellocyte production and differentiation. While the data in Figure 1 is quite compelling, I believe it important to test via sterile needle wound the wild-type and Or42a>Hid line. Alternatively, sterile needle wound alone may not be sufficient to trigger the heightened response, but only in combination with odorant. This may be the case as the authors examine a general injury response. However, the methods do not outline what a general injury response is from, so I cannot conclude the finding. Either way, this would be important to address.

As we understand here, the point being raised by the reviewer is about dampened lamellocyte formation in olfactory mutants is perhaps a consequence of the boulardi venom. The data we provide in the current manuscript throughout, clearly shows that cellular response to waspinfection in olfactory and GABA-metabolic mutants are specifically affected for lamellocyte formation while the overall blood cell numbers seen following infection are comparable to controls. This is reflected in comparable cell densities post wasp-infection between control *w1118* vs orco1/orco1, Orco> vs Orco>Hid, rpr, Or42a> vs Or42a>Hid, Kurs6> vs Kurs6>Gad1RNAi (Figure 1—figure supplement 1E). In all these conditions while the control makes a substantial percentage of lamellocytes, the mutants even though have similar cell densities they are incapacitated to generate lamellocytes. Implying that these animals respond to infections. The strength of wasp-infection induces heightened immune response as opposed to sterile injury as rightly mentioned byt the reviewer and clearly olfactory mutants and blocking progenitor GABA metabolic conditions are able to respond to infection but unable to mount lamellocyte differentiation. This is reflected in their lymph gland numbers and circulating lamellocyte numtbers. This defect is restored by GABA or succinate supplementation. Hence based on all the mutants described in the study, we feel it is very unlikely that all of them are more susceptible to the parasitoid venom leading to the lamellocyte defect.

– The defects in melanization yield a second important question: Are crystal cells also negatively affected by the inability to detect odorants? Are crystal cell populations affected by wasp odor? This question should be investigated and can be easily by heating *Drosophila* larvae as described in the citation below and counting the melanin spots. If this cell type is also affected, it would provide a stronger mechanistic link between lack of melanotic activity and odor detection of either both cell types OR only lamellocytes.– Crystal cells self-melanize when larvae are incubated at 60°C for 10 minutes– Williams, Ando and Hultmark, (2005).– This point is furthered by text in the manuscript: "GABA metabolism does not control differentiation of blood cells to plasmatocytes or crystal cell lineages, implying specificity of GABA in priming lamellocyte potential."

The current manuscript has been restructured as per the overall suggestions from the reviewers and the editor to center focus on lymph gland blood-progenitor cells and their contribution to lamellocyte formation. For this reason, we have omitted all the melanization data as we feel a clear understanding of this process requires much more experimental work and is better suited as an independent piece of work. As rightly pointed here in this comment as well, addressing crystal cells and their functions in response to wasp-odors in melanization response warrants further analysis, which is currently being undertaken.

– The odorant clearly primes the immune response of the *Drosophila* larvae. I am left wondering what is the odorant that does the priming. The Materials and methods read:

"L. boulardi wasps in the proportion of 15 females and 8 males into regular food medium".

– I believe it is important to the impact of the paper to ask whether the odorant detected is male wasp specific or female wasp specific (OR perhaps it is not specific?). Either way, this is an important outstanding question that should be addressed. Regardless of the answer, this will further catapult this exciting finding into becoming a seminal work in the field of environmental modulation of physiology. This will also provide a baseline to identify what exactly Or49a is detecting (male, female, or general wasp odor?). Pure male populations can be acquired by using virgin female wasps to infect larvae. All F1 wasps will be male, thus providing a pure odorant. I am excited to read future studies that will hopefully identify the molecule that is being detected.

We agree with the reviewer that it will be very exciting to identify the molecule that drives immue-priming. Based on data presented by (Ebrahim, Dweck et al., 2015) behavior of *Drosophila* larvae to wasps, it is mediated by Or49a. In this study the authors describe the specificity of Or49a towards detection of iridomyrmecin which is an odor produced specifically by *Leptopilina* wasps from both males and females. This study clearly shows the behavioral avoidance response to iridomyrmecin in the larvae is mediated by activation Or49a. Based on our genetic data with blocking Or49a function in WOF condition, we speculate iridomyrmecin as the molecule that functions in immune-priming capacity, however the commercial unavailability of this compound has limited us from asking this question and still remains to be tested.

Reviewer #3:This is a very interesting paper, throwing more light on the mysterious connection between olfaction and immunity, previously described by some of the authors of this manuscript. The data presented here show that olfactory detection of parasites, via one or two specific odorant receptors, is required to prime the immune system for an enhanced response to later parasite attacks. They also confirm that the signal from the central nervous system to the hematopoietic tissue is mediated by GABA. GABA is taken up by the blood cell precursors, affecting their cellular metabolism and stabilizing Sima, a homolog of hypoxia-inducible factor α (HIFalpha). There, Sima is required for the generation of immune response effector cells (lamellocytes). When the authors blocked olfactory signaling, for instance by mutating a key olfactory co-receptor, the animals were unable to make lamellocytes. This capacity could be rescued, for instance by directly providing GABA, or by genetically blocking Sima turnover, thereby increasing its concentration. The results are convincing, and the links described here between olfaction, HIF signaling and immunity should be of considerable general interest. However, the paper is not very well written and some important information is missing:

We sincerely thank reviewer#3 for the positive response of our work. We have made all attempts to improve the quality of the work both in terms of rigor and analysis by doing new experiments and in terms of re-structuring of the draft to provide important data that were missing in the previous version. Please find our response to every comment here with.

1) A very recent article by Krejčová et al., (2019) describes the role of HIF signaling in the activation of *Drosophila* blood cells during bacterial infection. That paper very nicely complements the results described here. It was perhaps published after this manuscript was submitted, but appropriate references to that article must be added.

We thank the reviewer for pointing this out. Yes, it was missed as our paper was submitted much before Krejčová et al., 2019 became online. This reference is now cited.

2) References are made to a "manuscript in submission" by Madhwal et al. Depending on how close that manuscript has come to publication, it may be wise to depend less on data presented there, since these data are still hidden from the reader. The authors could probably make their points by referring to other sources (e.g the above-mentioned paper by Krejčová et al.,), or to the experiments shown in the present manuscript.

We provide all the data on GABA and its metabolism in blood-progenitor cells in the revised version. The data is shown in Figure 2, Figure 3 and Figure 2—figure supplement 1, Figure 2—figure supplement 2, Figure 2—figure supplement 3, Figure 2—figure supplement 4, Figure 2—figure supplement 5, Figure 3—figure supplement 1, Figure 3—figure supplement, Figure 3—figure supplement 2.

3) Specifically, it is claimed that "Sima is both necessary and sufficient for lamellocyte induction". The data presented here suggest that Sima is necessary, and data elsewhere point in the same direction, but I am not aware of any published data showing that it is sufficient for lamellocyte induction. If that claim is only supported by the other submitted manuscript, it is better to delete it. The presented model does not depend on it.

We agree with the reviewer that Sima sufficiency is not supported by the current data we have. This is now corrected.

4) The experimental system is not fully described, leaving it to the reader to fill the gaps. For instance, it is not clearly stated which parasite is studied. The Introduction makes a general statement about "Leptopilina wasps", and in the later sections, the reader has to infer that "wasps" or "L. boulardi wasps" refers to wasps of the species Leptopilina boulardi. That may seem self-evident for people in the field, but to help others it should be stated explicitly.

We have made this clear in the text and describe the use of *L. boulardi* explicitly. The methods section incoporates a segment on wasp-infection to describe the experimental details with more clarity.

5) Many figures show the effects of genetic constructs, food etc. on lamellocyte production. From the context it can be inferred that these effects were often (maybe always?) studied in wasp-infected larvae. That must be clearly stated.

This is now stated clearly. For all data acquired upon infection, hours post infection is mentioned both in the figures and in legends and also this is carefully presented in the main text. Data acquired in uninfected conditions is also categorically described and we hope the current version will be clearer in this regard.

6) The second sentence in the Results section introduces "a subset of neurosecretory cells (Kurs6+)", implying that Kurs6+ is a term for a specific set of neurons. Later, "Kurs6" comes up as part of a genotype. That confused me at first. It took me some time to figure out that Kurs6-Gal4 is in fact a driver construct, and that "Kurs6+" simply refers to cells that express this driver. It would have been helpful to properly introduce this driver to the reader.

We have made this distinction on Kurs6 clearer in the text. We thank the reviewer for raising this point.

7) The term "iGABA" turned up rather unexpectedly in the text, and it confused me a lot. I don't think it is understood by the general reader. Does it simply refer to intracellular GABA? If so, I strongly suggest to spell it out, rather than introducing yet another multiple-letter combination. That would not make the text significantly longer.

This is also clearly stated in the current version.

[Editors’ note: what follows is the authors’ response to the second round of review.]

Essential revisions:1) Resistance to parasitoid waspThe authors provide an extremely important body of work. However, but the reviewers have a concern about the physiological significance of the phenotype. It is appropriate to hypothesize that an increase in lamellocyte production will yield a more potent immune response against parasitoids, as seen in other *Drosophila* species (i.e. D. suzukii). However, genetic perturbation that increase lamellocyte numbers, or perturbs the immune system in any manner, does not necessarily mean that the immune response mounted will be successful. The authors should provide experiments monitoring resistance to parasitoid wasps when the pathway they discovered is perturbated. There should monitor the impact of feeding larvae on WOF on resistance and how disturbing Or49A, Gat and Ssadh affect resistance to parasitoid wasp.

We thank the reviewers for their positive feedback on our manuscript.

The physiological significance of the increased lamellocyte phenotype on immune response mounted and whether it is successful or not has now been addressed by measuring the impact on wasp-egg encapsulation. This has been undertaken by assaying for: (1) encapsulation response (Vanha*-*aho Leena*-*Maija et al., 2015) and by measuring (2) percent melanization (Yang et al., 2015).

For encapsulation response, individual *Drosophila* larvae (60+12HPI) were sorted under stereomicroscope according to the presence or absence of black capsules. The number of encapsulated and un-encapsulated wasp-eggs per larvae were counted. The egg was scored as encapsulated when traces of melanin were found on it (as described in Vanha*-*aho Leena*-*Maija et al., 2015).

For percent melanization, individual infected *Drosophila* larvae (60+12HPI) were sorted under stereomicroscope according to the presence or absence of black capsules. Larvae without obvious black capsules were dissected to confirm whether they were infected. The number of larvae in the cohort that showed this melanization response was obtained as represented as the percetage larvae with melanization response to the total number of infected larvae (as described in Yang et al., 2015). The details of these assays have now been provided in the Materials and methods section.

Our results show that rearing *Drosophila* larvae on WOF (Figure 5A, B) or forced activation of Or49a (*Or49a>TrpA1*), (Figure 5R, S) that cause an increase in lamellocyte production, also show a significant increase in both encapsulation response (Figure 5T) and percent melanization (Figure 5—figure supplement 2A). Blocking the pathway, on the other hand in *Dome>Gat^RNAi^* BL29422 and *Dome>Ssadh^RNAi^* VDRC 106637KK (Figure 2P and Q), where a reduction in lamellocyte numbers was noticed, a dramatic reduction in encapsulation response and percent melanization is observed (Figure 5—figure supplement 2A-D). These results provide the physiological significance of the increased lamellocyte phenotype on effective wasp-egg clearance. Since encapsulation response of wasp-eggs requires concerted action of activated immune cells including plasmatocytes, crystal cells and lamellocytes (Dudzic et al., 2015, Anderl et al., 2016, Sorrentino et al., 2001), the implications of the overall improved encapsulation and melanization detected in WOF and *Or49a >TrpA1* could also imply an improved repertoire of activated immune cells in addition to increasing lamellocyte numbers.

With the addition of this new data, the importance of olfaction in immune priming is further strengthened. We sincerely thank the reviewers for bringing this point and these data are discussed in Results section.

2) RNAi effectivity and using one lineThe reviewers questioned the validity of the study as some results are based only one RNAi and their knockdown efficiencies were tested by using a ubiquitous and not in the actual tissues. They however recognize that the model is supported by the fact that they are testing different players affect the pathway. The reviewers however ask to repeat the experiments with Gat and Ssadh using another RNAi line to reinforce their conclusion.

We have now tested all RNAi lines available for both Gat and Ssadh in VDRC and Bloomington stock collection.

For Gat loss of function, we find that *UAS-Gat^RNAi^* GD (VDRC), showed a significant reduction in lamellocyte formation. This is evident at 48HPI in circulating population. At 24HPI any change in the lymph gland is however not detected. For Ssadh, we find driving Ssadh^RNAi^ BL55683 and Ssadh ^RNAi^ 14751/GD both led to mild reduction in lamellocyte formation. In the lymph gland the changes are significant as opposed to in the circulation. The new lines gave weaker responses but the data show trends that are comparable to the lines previously used in the study *UAS-Gat^RNAi^* BL29422 and *UAS-Ssadh^RNA^*^i^ KK. These data are provided in the Figure 2—figure supplement 2A and B.

3) Sima stainingFigure 3: There are discrepancies in the Sima staining which put question into the specificity of this staining/back ground. For example, some LGs showed a punctate expression of Sima in the posterior part of the LG (Figure 3F,G and H which is not seem in the other LGs).

The posterior expression of Sima protein pointed out by the reviewer in lymph gland tissues refers to Sima expression seen in the cells of the PSC (negative for Domeless expression and positively marked with antennapedia, a bonafied PSC marker). This data is now shown in Figure 3—figure supplement 1A, B.

In conditions Figure 3F, G, and H pointed out by the reviewer, a reduction in Sima protein expression in the Dome+ cells of MZ is seen. Hence, the expression in PSC cells (lacking Dome expression) in these lymph glands becomes readily evident. We provide supporting images in Figure 3—figure supplement 1M-T’, showing Domeless expression in these lymph glands to make this distinction clear.

Pictures in Figure 3B, K and m are not in agreement with quantifications in 3O. The same comment holds for Figure 3F-L and quantifications in J.

We provide multiple images that have been utilized for these quantifications as supporting data for the quantifications. These figures are appended along with this response (Author response image 1 and Author response image 1). We will be happy to add them as supplementary figures in the manuscript if need be. We have replaced the images in Figure 3G, I, L and M with better representative ones as well.

Expression of Sima in lamellocyte is also not convincing.

We now provide better representative images of lamellocytes at 12HPI and 24HPI to support Sima expression in lamellocytes. Data shown in Figure 3—figure supplement 1E-H. Compared to Sima protein levels seen in most cells of the lymph gland, its expression post-infection in Myospheroid positive cells is elevated albeit at different levels (Figure 3—figure supplement 1E-H). Some show very high expression, and some show moderately elevated Sima levels. This is seen both at 12HPI and 24HPI.

The specificity of the Sima antibody has to be checked.

The Sima antibody utilized in this study has been obtained from Prof. Utpal Banerjee’s lab where they have generated this antibody and confirmed its specificity in their own independent study (Wang et al., 2016). We also provide evidence to support its specificity in the lymph gland.

1) As reported earlier in Mukherjee et al., 2011, the Sima antibody used in this study shows comparable expression pattern in the lymph gland. We find basal Sima protein expression in all cells of the lymph gland with elevated expression detected in crystal cells (Figure 3A and Figure 3—figure supplement 1C-D’).

2) We have further confirmed the specificity of this antibody by staining for Sima protein in lymph glands expressing *sima^RNAi^* (*dome-MESO>UAS-sima^RNAi^*). We observe a significant reduction in Sima protein levels in this genetic condition. We provide this data in Figure 3—figure supplement 1V,V’ and Z.

Figure 3—figure supplement 1I is the difference in sima mRNA levels significant?

Yes, this is significant (***p<0.0001). P value added in Figure legend.

Overall, we have addressed all the above points raised regarding Sima protein expression pattern in the lymph gland, the specificity of Sima antibody and provide better representative images used for graphical representation of the data in Figure 3. More supporting images that comply with the Sima quantifications are provided along with this response to satisfy the concern with Sima quantifications (See Author response image and Author response image 2). If required, we will be happy to provide them as Figure supplements accompanying the main manuscript as well.

**Author response image 2. respfig2:** 

[Editors’ note: what follows is the authors’ response to the second round of review.]

Essential revisions:1) The regulatory cascade that goes from iGAbA to Ldh (Figure 6, left part) is not fully established, since several epistasis experiments are still lacking.For example, functional links in the lymph gland are not established between: (i) hph and sima, and more importantly between (ii) sima and Ldh and (iii) sima and iGABA. Epistasis experiments are lacking which precludes drawing in the model Figure 6 plain arrows representing functional connections.One key point concerns Sima functions. In the model, Sima is acting downstream of iGABA and is required for lamellocyte differentiation in response to wasp parasitism (Figure 6, left part). Unfortunately, these regulations are not yet definitively established in the ms.Why have authors not performed rescue experiments of DomeMESO>Gat RNAi or (Dome-MESO>Ssadh RNAi) of the mutant lymph gland phenotype by overexpressing Sima with the Dome-MESO gal4 driver? This is a key experiment that would establish whether Sima is the key payer downstream of iGaba.Concerning the functional link between Sima and lamellocyte differentiation: Does the overexpression of Sima with the dome-MESO gal4 driver lead to a cellular immune response similar to the one observed in response to wasp infection? These are key questions that have to be addressed to sustain the model proposed in Figure 6.

We understand the concern raised by the reviewer regarding the epistatic relationship between iGABA, sima and Ldh. We have made attempts to conduct these epistasis experiments but have not been successful. This is mainly due to technical limitation posed by over-expression of sima or Ldh using progenitor drivers *dome-MESO> or Tep4>* drivers which lead to larval lethality. As an alternative approach we undertook GABA and succinate supplementation experiments in control animals and evaluated Sima function. These supplemented animals showed elevated Sima protein expression in progenitor cells and when infected made more lamellocytes. Expressing *Sima^RNAi^* in progenitor cells, led to abrogation of the lamellocyte response seen with GABA or succinate supplementation. These data showed that GABA and Succinate mediated lamellocyte response is dependent on Sima function in progenitor cells. The data are shown in (Figure 4—figure supplement 1E). Although not conclusive the data are in agreement with our conclusions and we have now redone the model keeping in line with these concerns raised. We have used dotted lines to present the possibility of Sima and its metabolic function in lamellocyte differentiation as opposed to plain arrows representing functional connections.

Some experiments given in the manuscript for establishing functional links are irrelevant: this is the case of the epistasis experiment between oroc1 and Sima (via hph RNAi with the hml driver; sup Figure 4L-O). The hml-gal4 driver is not expressed at all in lymph gland progenitors where hph function is supposed to be required!

We have removed this figure. Our attempts to conduct the rescue of *orco^1^* mutants by expressing hph RNAi under the control of *Hml^Δ^Gal4* were mainly undertaken to provide an epistatic relationship between olfaction and lamellocyte formation in blood cells of the lymph gland. Doing the same experiment with progenitor specific drivers was not successful, mainly due to the genetic combinations of fly strains that had limited success surviving and were often unhealthy and failed to propagate. Hence, we utilized *Hml^Δ^Gal4* which is also a well-established lymph gland driver line, but yes as the reviewer points out it marks differentiating blood cells of the lymph gland. But using this driver as well we could recover lamellocyte formation in the lymph glands of *orco^1^* mutants. These data provided epistatic relationship of the systemic pathway in blood cells of the lymph gland and indicated that loss of *Hph* in Hml^+^ is also sufficient to restore their lamellocyte formation. But to minimize complexity, as suggested Figure 4—figure supplement 1L-O data are now eliminated.

2) Figure 6 (right panel) this representation is misleading and is not in agreement with the presented data. The proposed role for wasp odor for the immune response is not correct. Indeed, killing the Or49a neurons (wasp odor sensing neurons) has no impact on the immune response on larvae raised on RF (Figure 5). Thus, there is no need at all for these neurons to mount an immune response. Raising larvae on WOF leads only to an increase in the immune response that is dependent on Or49A neurons. Any conditions that lead to increased GABA levels (such as SF, GF,WOF) and even in the absence of wasp parasitism (since lamellocytes are detected in these conditions in the absence of parasitism) have the same consequence; i. e. a boost of the immune response. This data indicates that increasing GABA levels (by activating GABAnergic neurons) by different ways leads in all conditions to boosting the immune response.

We understand the point mentioned here and have now edited the model to mention “Boosting immune response”. We thank the reviewer for bringing this point to our attention.

3) Ca^2+^ levels in the lymph gland and their potential contribution to the immune response is unclear. What about Ca 2+ levels and requirement: (i) in response to wasp infection and (ii) in dome-MESO>gat RNAi conditions?

Ca^2+^ levels have been addressed and presented in the manuscript Figure 2—figure supplement 1P-R and Figure 2—figure supplement 3G, H.

pCamKII is calcium-calmodulin-dependent protein kinase II whose phosphorylation at Threonine286 is dependent on intracellular Ca^2+^ signaling (Miller et al., 1986). pCaMKII expression in the lymph gland progenitor cells has been shown to be responsive to GABA/GABA_B_R mediated signaling whose function downstream of GABA_B_R is necessary for progenitor maintenance (Shim et al., 2013). Hence, as readout of any change in progenitor Ca^2+^ signaling and GABA/GABA_B_R signaling, pCamKII expression analysis in lymph gland progenitor cells was conducted.

In control animals in response to wasp infection (6HPI) we did not observe any change in pCamKII levels, and the levels remained comparable to un-infected controls. This data is shown in Figure 2—figure supplement 1P-R.

In *dome-MESO>Gat^RNAi^* animals no change in pCamkII expression was detected and remained comparable to controls (Figure 2—figure supplement 3G, H). Based on this we concluded that the loss of lamellocyte differentiation in *Gat^RNAi^* animals is most likely independent of progenitor Ca^2+^ signaling. Also consistent with this result, expression of GABA_B_R^RNAi^ in progenitor cells, which has been shown to down-regulate progenitor intracellular Ca^2+^ levels (Shim et al., 2013) made lamellocyte formation. Hence, we understand that loss of Ca^2+^ may not impede lamellocyte formation but any definitive role for calcium signaling in infection response and lamellocyte development needs to be investigated more thoroughly and is best done as an independent study.

4) Figure 2—figure supplement 4 and Figure 4D-E, H-I controls are missing. This concerns Figure 4D and E; Figure 4H and I and Figure 2—figure supplement 4L.

The controls are provided in the figures. The graphs show rescue of lamellocyte phenotypes in *Gat* or *Ssadh* mutants, which lack succinate and *orco^1^* or *Kurs6>Gad1RNAi* conditions, which lack GABA and consequently succinate. When raised on succinate or GABA supplemented food, these mutants show rescue of lamellocytes formation comparable to controls raised on regular diet (Controls are indicated with red arrows in Author response image 3).

**Author response image 3. respfig3:** 

In response to wasp parasitism, do wild type larvae raised in SF or GF have increased lamellocyte differentiation compared to wt parasited larvae raised on RF?

This data is provided in the manuscript Figure 4—figure supplement 1E. Controls reared on SF or GF show increased lamellocyte differentiation.

5) Figure 2—figure supplement 3: Why are the authors looking at pCamKII ? What does this marker indicate?

As mentioned in the previous comment, pCaMKII expression in the lymph gland progenitor cells has been shown to be responsive to GABA/GABA_B_R mediated signaling whose function downstream of GABA_B_R is necessary for progenitor maintenance (Shim et al., 2013). Hence, as readout of any changes in progenitor Ca^2+^ signaling and GABA/GABA_B_R signaling, pCamKII expression analysis in lymph gland progenitor cells was conducted.

6) Figure 2—figure supplement 3P: DomeMESO>gatRNAi , a decrease in iGABA is observed (Figure 2H) whereas no difference for pxn is measured. This is different from data given in Shim et al., 2013 where a decrease in GABA leads to increase Pxn? How can one reconcile these data?

Shim et al., 2013 established a role for extracellular GABA (eGABA) function as a ligand, which activates GABA_B_R signaling in blood progenitors to sustain their intra-cellular calcium homeostasis necessary for their maintenance. Consequently, loss of GABA from the brain or GABA_B_R signaling in blood progenitors led to increased differentiation (both Pxn and P1) with a subsequent loss of MZ (medullary zone) maintenance markers.

The current study delineates an intracellular function for GABA (iGABA), whose internalization via GABA transporter (Gat) and breakdown via Ssadh is necessary for priming blood progenitor cells to differentiate into lamellocytes. GABA uptake by progenitor cells via Gat and its catabolism drives lamellocyte differentiation, which is independent of GABA’s function as an extracellular ligand driving GABA_B_R signaling. This is supported by Gat and Ssadh loss-of-function data that did not impede Ca^2+^ signaling in progenitor cells (supported by no change in pCamKII data shown in Figure 2—figure supplement 3G, H). Thus, progenitor GABA_B_R signaling is intact and hence their maintenance is unperturbed.

We conclude that neuronally derived extracellular GABA has dual function during blood progenitor development. (A) As a ligand, binding and activation of eGABA to progenitor GABA_B_R drives progenitor maintenance (Shim et al., 2013) and (B) eGABA internalization via Gat in progenitor cells and its breakdown establishes Sima protein expression essential for lamellocyte induction. We propose that the two pathways run parallel to each other and do not impede the functioning of each other. This is supported by data where *Gat* or *Ssadh* progenitor loss did not alter their GABA_B_R signaling and similarly, loss of GABA_B_R signaling in progenitor cells did not impede lamellocyte differentiation (Figure 2—figure supplement 1K and L) or Sima expression in progenitor cells (Figure 3—figure supplement 1P, P’ and Q) rather made more (Figure 2—figure supplement 1K).

7) Dome-MESO>sima RNAi: Subsection “Progenitor Sima protein stability via GABA-catabolism establishes lamellocyte Potential”: One cannot write that "plasmatocyte number is not affected" since quantifications are missing.

Now corrected.

8) Propositions for deleting some parts

– Figure 3—figure supplement 1M-T' are redundant with Figure 3.

These figures have now been removed.

– Data relative to tango: they are based on only one RNAi treatment and they are not essential. Thus, they could be removed.

We have removed the data.

– Figure 4A, B, C, F and G are not informative. Cells shown have a round shape and do not correspond to lamellocytes that have a characteristic elongated morphology.

These have now been removed.

– Figure 5—figure supplement 2G there is no reference of this Figure in the manuscript!

We have called this in Materials and method sections .

9) I have still comments on the writing as the article, which is difficult to follow even to quite close experts.

– The Abstract is difficult to understand:Here is a possible (suggested) Abstract based on your texts:

“Studies in different animal model systems have revealed the impact of odors on immune cells, However, any understanding on why and how odors control cellular immunity remained unclear. We find that *Drosophila* employ an olfactory/immune cross-talk to tune a specific cell type, the lamellocytes, from hematopoietic-progenitor cells. We show that neuronally released GABA derived upon olfactory stimulation, is utilized by blood-progenitor cells as a metabolite and through its catabolism, these cells stabilize Sima/HIFα protein in them. In blood-progenitor cells, Sima capacitates these cells with the ability to drive a metabolic state that is necessary for initiating lamellocyte differentiation. This systemic axis becomes relevant for larvae dwelling in wasp-infested environments where chances of infection are higher. By co-opting the olfactory route, the pre-conditioned animals elevate their systemic GABA levels leading to the up-regulation of blood-progenitor cell Sima expression. This elevates their immune-potential and primes them to respond rapidly when infected with parasitic wasps. The present work highlights the importance of the olfaction in immunity and shows how odor detection during animal development is utilized to establish a long-range axis in the control of immune-progenitor competency and priming.”Introduction “innate competitiveness” consider rewriting this term.

We have incorporated this suggested abstract and edited the Introduction accordingly.

Introduction: The third paragraph does not fit well the flow of the text. Include this text after mentioning the link between odor and immunity.

This is done.

Introduction: consider shortened this part or move some part in the Abstract.

This is now done.

IntroductionIndicates that you are using Orco and provide background information or replace by an introductory sentence if the result is described below.

This is now edited.

Indicate that it is in the “lymph gland”.

Done.

Replace mutational analysis by “the use of mutations” or “the use of loss of function mutations”.

Done.

Subsection “Olfaction controls cellular immune response necessary to combat parasitic wasp Infections”: Explain what is PSC and why you look at PSC.

We have added it in Subsection “Olfaction controls cellular immune response necessary to combat parasitic wasp Infections”:.

Subsection “Progenitor Sima protein stability via GABA-catabolism establishes lamellocyte Potential”: Avoid to use a passive sentence form. It will be simpler to say. We next explored…. (this applies to many other parts of the text).

This is corrected.

Consider to add additional sub-sectioning in the results to facilitate the reading.

We have added subheadings.

Subsection “Pathogenic odors induce immune priming”, second paragraph: I could not understand the statements. Could you make it clearer?

This is now re-written as “in *orco* mutant animals the WOF immune benefit was completely diminished (Figure 5—figure supplement 1A). These data showed that the WOF induced immune priming was not restricted to specific genetic backgrounds and secondly, it was mediated by olfactory stimulation and not mediated by feeding or ingestion of wasp-odor components.”

The Discussion is far too long and should be divided by two.

We have shortened it.